# REINFORCEMENT LEARNING FOR EVIDENCE-SEEKING DIAGNOSTIC REASONING WITH LARGE LANGUAGE MODELS

## ABSTRACT

Recent large language models (LLMs) excel at reasoning but often assume complete information, whereas real-world tasks, such as medical diagnosis, require iterative collections of evidence. Existing research rarely reflects this process, treating diagnosis as a one-turn task. This work explicitly formalizes this as a two-turn diagnostic paradigm and proposes reinforcement learning with diagnostic evidence-seeking rewards to guide LLMs in requesting and using evidence. We further introduce Retrieval-Augmented Generation-based Examination Simulation (RAGES), which generates realistic and plausible follow-up evidence to facilitate the process. Experiments on multilingual datasets show that (1) LLMs significantly improve diagnostic accuracy with additional evidence, (2) our model outperforms or matches larger and reasoning-enhanced baselines, and (3) RAGES generates more plausible results than pure LLM generation.

## 1 INTRODUCTION

With the remarkable performance of models such as OpenAI o1 (Jaech et al., 2024) and DeepSeek-R1 (DeepSeek-AI, 2025), the intrinsic reasoning capabilities of large language models (LLMs) have attracted increasing research attention. However, in real-world scenarios, gathering information to support reasoning is inherently an iterative and accumulative process. LLMs may not provide definitive answers at early stages when reasoning is based on limited evidence. In such situations, the ability to request additional evidence becomes crucial.

This scenario is common in the medical domain, where evidence-based diagnosis is essential (Emmi et al., 2023). Clinicians rarely have complete information initially and must actively gather more evidence through patient interactions, additional tests, and inter-departmental consultations. Pathological diagnosis typically follows a two-step process. Pathologists first generate candidate diseases based on preliminary evidence, then order further tests to refine hypotheses into a final diagnosis. For LLMs, managing such requests is non-trivial, as it requires multiple rounds of interaction.

Despite its clinical significance, prior LLM-based diagnostic research has mainly focused on single-turn, multiple-choice tasks that assume complete information is readily available (Qiu et al., 2025; Zhang et al., 2025). Such settings overlook the critical challenge of reasoning under uncertainty and the need to actively seek evidence. Furthermore, evaluating open-ended differential diagnoses is inherently difficult due to their subjective nature. Even experts may generate different disease hypotheses and recommend varying tests for the same case, making this task particularly challenging.

To address these challenges, we propose the use of **reinforcement learning with verifiable rewards** (DeepSeek-AI, 2025; Mroueh, 2025), tailored to an evidence-seeking diagnostic setting. To handle the open-ended nature of this task, LLMs act as auxiliary judges (Guan et al., 2024). We design three complementary reward signals: the **format reward**, which enforces structured and extractable outputs; the **rank-sensitive diagnosis reward**, which encourages broad yet clinically meaningful differential diagnoses; and the **examination consistency reward**, which ensures that recommended follow-up tests are both plausible and aligned with suspected diseases. We also introduce **Retrieval-Augmented Generation-based Examination Simulation** (RAGES), which generates realistic follow-up evidence based on LLM queries. Empirical results show that LLMs achieve substantial diagnostic accuracy when provided with further evidence, and our models consistently

outperform or match strong baselines. Moreover, RAGES produces more reliable and clinically plausible follow-up evidence than raw LLM outputs alone.

Our contributions are threefold: (1) We demonstrate that LLMs achieve substantially higher diagnostic accuracy when provided with additional evidence, highlighting the importance of iterative evidence gathering in medical diagnosis. (2) We develop a reinforcement learning framework with verifiable, diagnostic evidence-seeking rewards that explicitly encourage evidence-based reasoning in diagnosis. (3) We introduce RAGES, a mechanism that generates realistic and clinically plausible follow-up evidence in response to LLM queries.

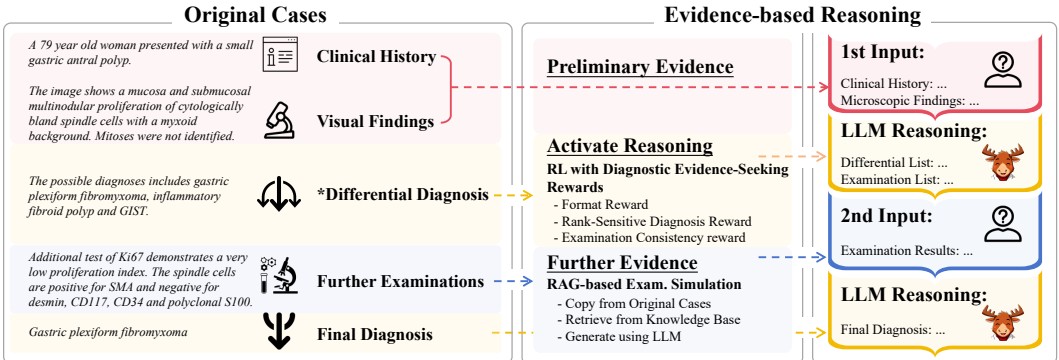

Figure 1: An overview of this work. **Left:** We collect case reports and extract five aspects of information. **Right:** We adopt an evidence-based, two-turn reasoning process regarding diagnosis. To realize this process, we utilize RL with diagnostic rewards to activate the reasoning capability, and propose an RAG-based method to simulate the test results as further evidence.

## 2 RELATED WORK

### 2.1 REASONING CAPABILITIES OF LLMS IN MEDICAL DIAGNOSIS

Early studies have shown that pretrained LLMs encode rich clinical knowledge and can effectively answer medical questions, for example, Flan-PaLM (Singhal et al., 2023) and MedFound (Liu et al., 2025c). Prompting techniques such as chain-of-thought (CoT) have proven effective for inducing reasoning (Wei et al., 2023; Besta et al., 2024; Yao et al., 2023). In medicine, structured prompting enhances diagnostic accuracy (Nori et al., 2023; Savage et al., 2023; Kwon et al., 2024; Savage et al., 2024). With the advent of OpenAI's o1 model (Jaech et al., 2024), the focus shifts to the native reasoning capability of LLMs on medical tasks. Nori et al. (2024) evaluate o1-preview on medical challenge problems and find it dramatically outperforms previous models with prompting. Sandmann et al. (2025) and Tordjman et al. (2025) both evaluate DeepSeek-R1 (DeepSeek-AI, 2025) on medical tasks and clinical reasoning, demonstrating the potential of reasoning models. Building on this new paradigm, recent work has introduced medical LLMs and frameworks designed for stepwise reasoning. HuatuoGPT-o1 (Chen et al., 2024) is a medical LLM trained via verifiable reasoning steps. Huang et al. (2025) focus on inference-time scaling of reasoning in the medical domain. MedS³ (Jiang et al., 2025a) learns to reason about medical problems with a process reward model. Baichuan-M2 (Team et al., 2025) has shown excellent reasoning capability in medical tasks.

### 2.2 REINFORCEMENT LEARNING WITH VERIFIABLE REWARDS

Reinforcement learning for LLMs originates with reinforcement learning from human feedback (RLHF), which is introduced to better align model outputs with human preferences. More recently, with the advent of DeepSeek-R1 and group relative policy optimization (GRPO) framework, more attention has shifted toward rule-based reward functions as a means to elicit and strengthen the intrinsic reasoning capabilities of LLMs (Mroueh, 2025). In mathematics, functions primarily focus on correctness verification, as seen in Open-R1 (Hugging Face, 2025) and DeepScaleR (Luo et al., 2025). In the medical domain, several studies utilize closed-ended questions and also employ direct

correctness verification (Qiu et al., 2025; Zhang et al., 2025; Liu et al., 2025a; Tarek & Beheshti, 2025; Zhu et al., 2025). However, for tasks where answers cannot be easily verified, LLM-as-a-judge has emerged as a practical alternative for reward evaluation (Guan et al., 2024).

### 2.3 MULTI-TURN DIAGNOSIS

Clinical diagnosis is inherently iterative, involving hypothesis formation, information gathering, and refinement. Several studies simulate multi-turn doctor-patient interactions (Bao et al., 2023; Chen et al., 2023; Li et al., 2023; Toma et al., 2023; Liu et al., 2025d), including systems such as AMIE (Tu et al., 2024), AI Hospital (Fan et al., 2025), and MedAgentSim (Almansoori et al., 2025). APP (Zhu & Wu, 2025) explores patient-centered multi-turn consultations, while MedAgentBench (Jiang et al., 2025b) and MMD-Eval (Liu et al., 2025b) provide realistic simulation environments grounded in structured patient data. Other efforts focus on sequential stages of diagnosis. Sun et al. (2024) note that most LLM-based studies treat diagnosis as one-shot dialogue and propose a two-planner system for differential diagnosis and final prediction. Similarly, MAC (Chen et al., 2025) models two consultation stages and simulates multidisciplinary treatment. However, few studies tackle continuous multi-turn reasoning with additive evidence within a single LLM session.

## 3 METHOD

### 3.1 EVIDENCE-SEEKING REASONING

We formalize the diagnostic workflow as a two-turn, evidence-seeking interaction with LLMs, as illustrated on the right side of Fig. 1. In the first turn, the model receives initial information and generates a set of candidate hypotheses along with suggestions for additional tests. In the second turn, the model incorporates the simulated results of these follow-up tests, refines its reasoning, and outputs a final diagnosis. This two-turn paradigm is established with three key components. First, we utilize real-world diagnostic cases collected from multiple sources to provide plentiful information for learning (Section 3.4). Second, our proposed RL framework trains the model to produce reasonable differential diagnoses and to actively request further evidence (Section 3.2). Third, the RAGES method connects different turns by generating simulated test results to guide subsequent reasoning (Section 3.3). This setup provides a principled framework for studying evidence-seeking behavior and the efficacy of diagnostic reasoning in LLMs.

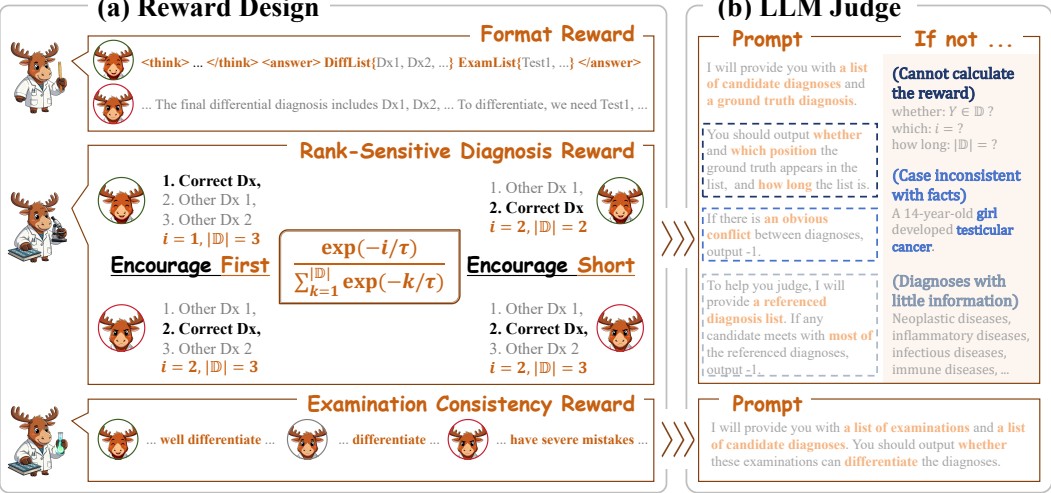

Figure 2: The proposed RL with diagnostic evidence-seeking rewards. (a) The reward design. (b) The prompts used in the process of LLM judging.

## 3.2 REINFORCEMENT LEARNING WITH DIAGNOSTIC EVIDENCE-SEEKING REWARDS

To elicit the intrinsic reasoning capabilities of LLMs, we employ reinforcement learning with verifiable rewards within the GRPO framework, utilizing the diagnostic evidence-seeking rewards. Given the complexity of evaluating diagnostic outputs, we leverage LLMs as judges to assess both whether a predicted diagnosis aligns with the ground truth and whether the proposed examinations are sufficient for discriminating disease candidates.

### 3.2.1 DIAGNOSTIC EVIDENCE-SEEKING REWARDS

The reward function comprises three complementary components: format reward, rank-sensitive diagnosis reward, and examination consistency reward. The **format reward** enforces a clear separation between reasoning and answer, ensuring structured responses that facilitate the extraction of target outputs. The **rank-sensitive diagnosis reward** $R_d$ is designed to emphasize the primary diagnosis at the top of the candidate list while shortening the candidate list if necessary. The **examination consistency reward** $R_e$ provides a bonus for proposing clinically plausible follow-up tests. The overall reward function is defined as:

$$R(X, Y) = \begin{cases} R_d(\mathbb{D}, Y) + R_e(\mathbb{T}, \mathbb{D}) - P_h \times \mathbf{1}_c & n_f = 0, \\ -P_f \times n_f & n_f \neq 0, \end{cases} \tag{1}$$

where $X$ is the model output, and $Y$ the ground truth. $n_f \in \{0, 1, 2\}$ denotes the number of format errors, including missing think/answer pairs or improperly presented answers. $\mathbb{D}$ represents the extracted ordered diagnosis list, and $\mathbb{T}$ the test list. $P_f$ and $P_h$ denote the format and the hacking penalty, respectively. $\mathbf{1}_c$ indicates the occurrence of hacking, which is discussed in Section 3.2.2.

The examination consistency reward $R_e$ assisted by an LLM judge is defined as,

$$R_e(\mathbb{T}, \mathbb{D}) = \begin{cases} B_e & \mathbb{T} \text{ can differentiate } \mathbb{D} \text{ effectively,} \\ 0 & \mathbb{T} \text{ have no severe conflicts with } \mathbb{D}, \\ -B_e & \mathbb{T} \text{ contain some severe problems.} \end{cases} \tag{2}$$

where $B_e$ is the value of the bonus.

The rank-sensitive diagnosis reward $R_d$ is defined as,

$$R_d(\mathbb{D}, Y) = \begin{cases} \frac{e^{-i/\tau}}{\sum_{j=1}^{|\mathbb{D}|} e^{-j/\tau}} & Y = D_i \in \mathbb{D}, \\ 0 & Y \notin \mathbb{D}, \end{cases} \tag{3}$$

where $\tau > 0$ is a hyperparameter controlling the score distribution. The same LLM is used to determine whether and at which position the proposed diagnoses align with the ground truth.

The reward $R_d$ exhibits some features: (1) pushing primary diagnoses to appear first (**encouraging first**), (2) favoring shorter candidate lists (**encouraging short**), (3) adaptively influencing the accuracy of the main diagnosis, and (4) affecting the length of the diagnosis list. The first two can be derived from the definition, as in Theorems 1 and 2. The latter two are more complex, so we refer to them as Findings 1 and 2 with simple theoretical analysis. Detailed proofs are in Appendix A.2. To more intuitively illustrate these properties, Fig. 3 visualizes three representative charts. The left panel instantiates the rank-sensitive diagnosis rewards at different hit positions across varying list lengths under a fixed $\tau$, highlighting the first two properties. The middle panel illustrates the first finding by comparing different choices of $\tau$ with a fixed list length. The right panel shows the reward gains obtained when the list length decreases from 5 to 4 across different $\tau$, consistent with the second finding.

**Theorem 1** The rank-sensitive diagnosis reward confers a higher value when the correct diagnosis appears earlier in the diagnosis list. More formally, consider two diagnosis lists of equal length ($|\mathbb{D}^1| = |\mathbb{D}^2|$) that both contain the ground truth diagnosis $Y$. Let the position of the correct

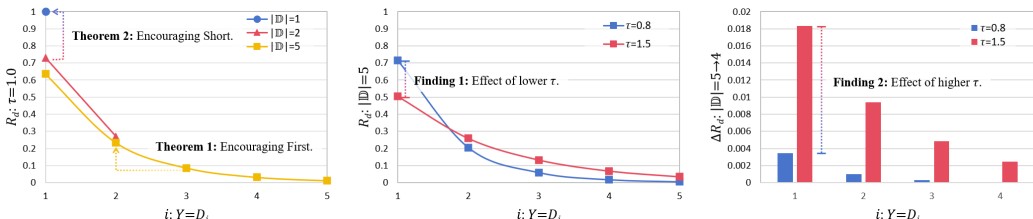

Figure 3: Visualization of the proposed rank-sensitive diagnosis reward. **Left**: the rewards of different positions under various diagnosis list lengths with a fixed $\tau = 1.0$; **Middle**: the rewards under a fixed length $|\mathbb{D}| = 5$ with different $\tau$; **Right**: the reward gains when the length shortens as $|\mathbb{D}| = 5 \to 4$ with different $\tau$.

---

diagnosis in $\mathbb{D}^1$ be $i$ and in $\mathbb{D}^2$ be $j$, such that $D_i^1 = Y$ and $D_j^2 = Y$ with $1 \leq i < j \leq |\mathbb{D}^1| = |\mathbb{D}^2|$. In this case, $R_d(\mathbb{D}^1, Y) > R_d(\mathbb{D}^2, Y) > 0$.

---

**Theorem 2** The rank-sensitive diagnosis reward assigns a higher value when the diagnosis list is relatively short. More formally, consider two diagnosis lists $\mathbb{D}^1$ and $\mathbb{D}^2$ of different lengths $(|\mathbb{D}^1| < |\mathbb{D}^2|)$ that both contain the ground truth diagnosis $Y$. Let the position of the correct diagnosis be $i$ in both lists, such that $D_i^1 = Y$ and $D_i^2 = Y$ with $1 \leq i \leq |\mathbb{D}^1| < |\mathbb{D}^2|$. In this case, $R_d(\mathbb{D}^1, Y) > R_d(\mathbb{D}^2, Y) > 0$.

Theorem 1 reflects one of our design goals: to mimic the behavior of doctors by prioritizing diseases with higher probabilities at the top of the differential list. Theorem 2 captures another desirable property that it implicitly prevents the differential list from becoming excessively long.

**Finding 1** A lower $\tau$ in the rank-sensitive diagnosis reward encourages the correct diagnosis to the top of the diagnosis list. More formally, consider a fixed diagnosis list $\mathbb{D}$ that contains the ground truth diagnosis $Y$, where the position of $Y$ is 1 (i.e., the top-1 diagnosis). For two hyperparameters satisfying $\tau_1 < \tau_2$, we have $R_d^{\tau_1} \geq R_d^{\tau_2}$. The equality holds if and only if $|\mathbb{D}| = 1$.

**Finding 2** A higher $\tau$ encourages the generation of shorter diagnosis lists. More formally, consider two diagnosis lists of differing lengths, where $|\mathbb{D}^1| < |\mathbb{D}^2|$, and let the correct diagnosis $Y$ appear at position $i$ in both lists. Define the reward increment with respect to $\tau$ as $\Delta R_d^\tau = R_d^\tau(\mathbb{D}^1, Y) - R_d^\tau(\mathbb{D}^2, Y)$. For hyperparameters satisfying $\tau_1 > \tau_2$, we have $\Delta R_d^{\tau_1} > \Delta R_d^{\tau_2} > 0$.

Formulating how the reward function "encourages" the observed phenomena is nontrivial. Hence, we present informal findings and provide a static analysis. Finding 1 shows that with a lower $\tau$, a correct top-1 diagnosis receives a larger reward. When $\tau \to 0$, the reward for the correct top-1 diagnosis approaches $R_d \to 1$, while correct diagnoses in other positions receive $R_d \to 0$. Finding 2 indicates that a larger $\tau$ increases the reward for shortening the diagnosis list. When $\tau \to \infty$, the reward for a correct diagnosis becomes uniform across all positions, $R_d = 1/|\mathbb{D}|$.

Motivated by these findings, we adopt a dynamic $\tau$ strategy: a lower $\tau$ is used when only preliminary information is available, encouraging the model to generate longer differential lists while emphasizing leading candidates. A higher $\tau$ is applied when additional information, such as IHC results, is available, encouraging the model to produce precise diagnoses with sufficient detail.

### 3.2.2 PENALIZING HACKING BEHAVIORS

Unlike mathematical problems with well-established normalization methods, medical diagnosis is inherently more complex. To address this, we leverage **LLMs as judges** to evaluate two key aspects: whether a proposed diagnosis aligns with the ground truth, and whether recommended examinations are appropriate and sufficient. However, using LLMs in this role introduces potential vulnerabilities

to **hacking behaviors** (Tarek & Beheshti, 2025). As shown in Fig. 2(b), we identify two main challenges: counterfactual diagnoses and diagnoses with insufficient information.

The first challenge arises from corner cases in the training data. For instance, models may systematically append *lymphoma* to differential lists, or a female patient could be erroneously suspected of having *testicular cancer*. The second challenge stems from evaluation ambiguities. LLMs may favor vague diagnostic descriptions that can match many diseases, thereby artificially inflating rewards.

To mitigate these issues, we introduce a hacking penalty term $P_h$ and improve the diagnosis judgment prompt with two mechanisms: conflict detection and differential-as-mirror. **Conflict detection** ensures consistency with the ground truth and basic medical knowledge, triggering the penalty if diagnoses contain factual errors or contradictions. **Differential-as-mirror** leverages referenced differential diagnoses from original cases. These diagnoses serve as closely related but wrong examples. If a diagnosis matches most of these negative samples, it is deemed overly vague and clinically uninformative, triggering the hacking penalty. Consequently, the final reward function is defined in Equation 1, where the condition c holds if any hacking occurs, resulting in a hit position $i = -1$.

### 3.3 RAG-based Examination Simulation

To complete the evidence-seeking process of request, acquisition, and utilization, it is necessary to propose a method of examination simulation. In real-world practice, further requested results can be obtained through pathologists or laboratory tests, enabling multi-turn interactions. During training, however, direct consultation with laboratories is impractical. To address this, we propose **Retrieval-Augmented Generation-based Examination Simulation** (RAGES), as outlined in Algorithm 1.

---

**Algorithm 1:** RAGES

**Input:** Case report $\mathcal{C}$, requested exams $E$, structured knowledge base $\mathcal{K}$
**Output:** Simulated examination outputs $\mathcal{E}_{\text{gen}}$
$(\mathcal{E}_{\text{gt}}, \mathcal{D}_{\text{gt}}) \leftarrow \text{ExtractInfo}(\mathcal{C})$ `# Get real exams and ground truth diagnosis`
$\mathcal{E}_{\text{direct}} \leftarrow \text{MatchOverlap}(\mathcal{E}_{\text{gt}}, E)$ `# Reuse overlapping real results`
$\text{Candidates} \leftarrow \text{EmbedAndSearch}(\mathcal{K}, \mathcal{D}_{\text{gt}})$
$\text{BestMatch} \leftarrow \text{SelectHighestSimilarity}(\text{Candidates})$
$\mathcal{E}_{\text{retrieved}} \leftarrow \text{GetMappings}(\mathcal{K}, \text{BestMatch})$ `# Retrieve disease-exam mappings`
$\mathcal{E}_{\text{gen}} \leftarrow \text{LLMGenerate}(E, \mathcal{E}_{\text{direct}}, \mathcal{E}_{\text{retrieved}}, \mathcal{D}_{\text{gt}})$ `# Generate final results via LLM`
**return** $\mathcal{E}_{gen}$

---

**Reuse of Existing Results.** For each case, we first identify overlaps between the examinations ordered by the model and those already performed in the record. Any overlapping results are directly reused, as they originate from verified laboratory data. However, considering the difference between the examination items, this step typically contributes only a portion of the desired results.

**Retrieval from Structured Knowledge.** To handle examinations not mentioned in the case report, we query a curated database containing over 24,000 mappings between 1,629 diseases and 465 IHC markers. Each mapping encodes statistical associations between diseases and test results. We embed the case's final diagnosis using a sentence transformer and search for the closest disease entries in the database. Its associated mappings are retrieved to augment result generation.

**LLM-based Generation.** Finally, we prompt a powerful LLM with the case context and retrieved knowledge to generate plausible outputs with the LLM's internal knowledge. By combining reused results, retrieved mappings, and LLM generation, RAGES prioritizes high-quality, explainable, and complete outputs while minimizing noise from hallucinations or overconfident LLM predictions.

### 3.4 Data Curation

Detailed case reports form the foundation of our evidence-based two-turn diagnostic process. Yet, pathology-specific datasets remain limited. To address this, we compiled cases from publicly available sources, including academic journals and medical websites. To make these cases usable for interaction with LLMs, we employed LLMs to extract structured information. Specifically, each case is broken down into five components, as shown on the left side of Fig. 1: clinical history,

microscopic findings, differential diagnosis, further examinations with their results, and the final diagnosis. The first two components provide the initial input for the first turn. The differential diagnosis is used to aid LLM judges, while the examination results offer dependable evidence for constructing the second-turn input. Finally, the confirmed diagnosis is treated as the ground truth for evaluating model predictions.

# 4 EXPERIMENTS

## 4.1 IMPLEMENTATION

We use GPT-4 to extract key information from the raw case reports and employ DeepSeek-R1 (DeepSeek-AI, 2025) to perform RAGES. For model training, we adopt Qwen2.5-7B-Instruct (Yang et al., 2024) as the base model and Qwen2.5-32B-Instruct as the LLM judge. The training dataset comprises 959 instances collected from DakaPath[1] and the Chinese Journal of Pathology, of which 287 contain real test results and 384 provide a referenced differential list. We use $\tau_1 = 0.8$ for cases with only limited information and $\tau_2 = 1.5$ for cases with additional test results. $P_f$, $P_h$, and $B_e$ are set to 0.5, 0.3, and 0.1, respectively. The training workflow is implemented using OpenRLHF (Hu et al., 2024) and conducted on 8 H100 GPUs, with four GPUs dedicated to model training and the remaining four to LLM-based judging. The training process completes in about 40 hours, which is dicussed in Appendix A.6. Additional details of the dataset and training settings are provided in the Appendix A.9 and A.10.

## 4.2 EVALUATION CONFIGURATIONS

### 4.2.1 EVALUATION DATASETS

To ensure fair and rigorous evaluation, we curated cases from multiple sources other than the training sources, including publicly available English-language datasets from **Pathology Outlines**[2] and the **Hans Popper Hepatopathology Society** (**HPHS**)[3], as well as **in-house Chinese-language cases** used for resident training at Hospital X. Relevant diagnostic information was manually extracted. We compiled two subsets. The **Public English Dataset** (EN) comprises 110 cases, of which 100 are from Pathology Outlines and 10 from HPHS. Due to data-sharing constraints, we release only the URLs for these cases. The **In-house Chinese Dataset** (CN) consists of 276 cases from Hospital X. Since our models are trained exclusively on Chinese data, the English dataset provides an unbiased and open-source benchmark. However, it primarily consists of complex or atypical cases. The Chinese dataset offers a more realistic setting while remaining challenging for evaluation purposes.

### 4.2.2 EVALUATION METRICS

For the initial consultation stage (*Initial*), where the model proposes potential disease candidates, we evaluate whether the ground truth diagnosis appears in the differential list. A match is considered a hit, and the hit rate quantifies the differential accuracy (*DiffAcc*). For the follow-up stage (*Follow-up*), where the model outputs a precise diagnosis, we check whether the ground truth appears as the top-ranked candidate. The hit-at-one rate is used to measure diagnostic accuracy (*DxAcc*). To ensure a comprehensive and objective evaluation, we leverage three stronger LLMs from different sources than what we deployed during training as automatic verifiers to migrate hacking problem through merely model-to-model agreement: DeepSeek-R1 (R1), Qwen2.5-Max (QM), and GPT-5-Minimal (GPT-5). Prior work (McDuff et al., 2025) has demonstrated that LLM-based evaluation shows high consistency with human experts. We also report the average score across these models.

### 4.2.3 EVALUATION BASELINES

In addition to our RL-based model (*Ours-RL-7B*), we distill reasoning data from DeepSeek-R1 and perform supervised fine-tuning of Qwen2.5-32B-Instruct (*Ours-SFT-32B*) following Huang et al. (2025). Details of the SFT process are provided in Appendix A.8. We also compare against

---

[1] https://www.dakapath.com

[2] https://www.pathologyoutlines.com

[3] https://hanspopperhepatopathologysociety.org

the following baselines: the original Qwen2.5-32B-Instruct (*Qwen2.5-32B*), a larger Qwen2.5-72B-Instruct (*Qwen2.5-72B*), two reasoning-enhanced models, *QwQ-32B* (QwenTeam, 2025) and *Qwen3-32B* (Team, 2025), and two medical reasoning models, HuatuoGPT-o1-7B (*Huatuo-7B*) and Baichuan-M2-32B (*M2-32B*).

# 5 RESULTS ANALYSIS

## 5.1 MORE EVIDENCE, MORE ACCURATE DIAGNOSIS

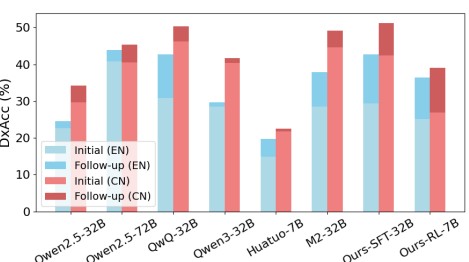
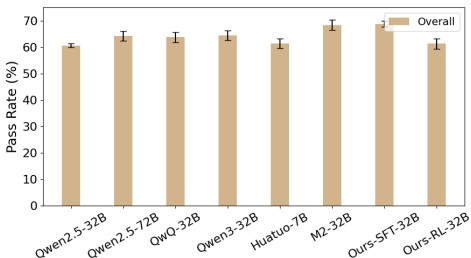

Figure 4: The comparison of diagnosis accuracy in the initial and follow-up consultation.

Figure 5: The plausibility of examinations requested by different models.

Before presenting a comprehensive comparison of two-turn diagnostic performance, we first examine whether LLMs produce more accurate diagnoses when provided with additional evidence. Specifically, we compare diagnostic accuracy after the initial consultation, which represents the traditional one-turn diagnosis, versus the follow-up stage, which represents the proposed evidence-seeking diagnosis. As shown in Fig. 4, all models achieve higher diagnostic accuracy on both datasets when follow-up information is available. While this result may appear intuitive, it provides a critical validation for our study: incremental evidence substantially improves LLM-assisted diagnosis, thereby motivating the design of the subsequent two-turn evidence-seeking evaluation and analyses. Moreover, since the cases might contain some further examination results, we present an idealized experiment on one-turn diagnosis with more evidence in Appendix A.3.

## 5.2 PERFORMANCES IN TWO-TURN DIAGNOSIS

Tables 1 and 2 summarize model performance across both the initial and follow-up consultation stages. For clarity, we also report the performance gains of our models relative to Qwen2.5-32B. During the initial consultation, both the RL and SFT models achieve strong performance, comparable to or exceeding the baselines. On the English dataset, the RL model exhibits a larger improvement over models with intrinsic reasoning, highlighting the effectiveness of RL in enhancing differential accuracy across multilingual settings. For diagnostic accuracy in the follow-up stage, the RL model does not fully match the performance of larger reasoning-enhanced models, though it still outperforms Qwen2.5-32B and a medical reasoning model Huatuo-7B. This may be due to the RL model not being directly trained with a definite-diagnosis objective for the rule-out process. Although we use different $\tau$ values to modulate the adaptive diagnostic reward, the resulting signal may be insufficient to fully optimize performance in a 7B parameter model. In contrast, the SFT model demonstrates the power of reasoning, achieving excellent performance relative to all baselines. Notably, Huatuo-7B can barely follow the required format and switches to another fixed format, so we report performance under that format; otherwise, the results would be almost 0.

## 5.3 THE PLAUSIBILITY OF EXAMINATIONS

To evaluate the plausibility of follow-up examinations proposed by each model during the initial consultation, we use GPT-5-Minimal as an external evaluator. GPT-5 is tasked with assessing whether the suggested examination items are appropriate given the differential diagnosis generated by the model. To ensure stability and reduce variance, each evaluation is repeated three times, and the pass

Table 1: The differential accuracy (DiffAcc) of models in the initial consultation (Yellow: chat models; Red: reasoning models). **Bold** figures suggest the best performance, and the underlined are the second best. Δ suggests the gains over the original Qwen2.5-32B.

| Model | Public English Dataset | | | | In-house Chinese Dataset | | | | Overall |
|---|---|---|---|---|---|---|---|---|---|
| | R1 | QM | GPT-5 | Avg. | R1 | QM | GPT-5 | Avg. | |
| Qwen2.5-32B | 42.7 | 43.6 | 41.8 | 42.7 | 51.8 | 54.7 | 49.3 | 51.9 | 49.3 |
| Qwen2.5-72B | 55.5 | 53.6 | 53.6 | 54.2 | 64.9 | 67.8 | 64.1 | 65.6 | 62.4 |
| QwQ-32B | 56.4 | 61.8 | 54.5 | 57.6 | 66.3 | 68.5 | 63.4 | 66.1 | 63.6 |
| Qwen3-32B | 40.9 | 46.4 | 45.5 | 44.3 | 55.4 | 59.1 | 54.3 | 56.3 | 52.8 |
| *Medical Reasoning Models* | | | | | | | | | |
| Huatuo-7B | 27.3 | 34.5 | 32.7 | 31.5 | 37.0 | 41.7 | 41.7 | 40.1 | 37.7 |
| M2-32B | 53.6 | 54.5 | 50.0 | 52.7 | 65.9 | 68.1 | 64.1 | 66.0 | 62.2 |
| *Our Models* | | | | | | | | | |
| **Ours-SFT-32B** | 55.5 | 57.3 | 51.8 | 54.9 | **67.0** | **69.6** | **64.5** | **67.0** | 63.6 |
| Δ | 12.8 | 13.7 | 10.0 | 12.2 | 15.2 | 14.9 | 15.2 | 15.1 | 14.3 |
| **Ours-RL-7B** | **60.9** | **68.8** | **56.4** | **62.0** | **67.0** | 68.5 | 62.0 | 65.8 | **64.8** |
| Δ | 18.2 | 25.2 | 14.6 | 19.3 | 15.2 | 13.8 | 12.7 | 13.9 | 15.4 |

Table 2: The diagnosis accuracy (DxAcc) of models in the follow-up consultation (Yellow: chat models; Red: reasoning models). **Bold** figures suggest the best performance, and the underlined are the second best. Δ suggests the gains over the original Qwen2.5-32B.

| Model | Public English Dataset | | | | In-house Chinese Dataset | | | | Overall |
|---|---|---|---|---|---|---|---|---|---|
| | R1 | QM | GPT-5 | Avg. | R1 | QM | GPT-5 | Avg. | |
| Qwen2.5-32B | 24.5 | 30.9 | 18.2 | 24.5 | 36.6 | 37.7 | 28.3 | 34.2 | 31.4 |
| Qwen2.5-72B | 41.8 | **50.9** | 39.1 | **43.9** | 46.7 | 50.0 | 39.5 | 45.4 | 45.0 |
| QwQ-32B | **44.5** | 42.7 | **40.9** | 42.7 | 51.8 | 52.2 | 47.1 | 50.4 | 48.2 |
| Qwen3-32B | 33.6 | 26.4 | 29.1 | 29.7 | 45.7 | 42.0 | 41.7 | 43.1 | 39.3 |
| *Medical Reasoning Models* | | | | | | | | | |
| Huatuo-7B | 23.6 | 18.2 | 17.3 | 19.7 | 23.9 | 22.8 | 20.7 | 22.5 | 21.7 |
| M2-32B | 40.0 | 35.5 | 38.2 | 37.9 | 53.6 | 46.4 | **47.5** | 49.2 | 46.0 |
| *Our Models* | | | | | | | | | |
| **Ours-SFT-32B** | 43.6 | 45.5 | 39.1 | 42.7 | **54.3** | **53.6** | 46.0 | **51.3** | **48.9** |
| Δ | 19.1 | 14.6 | 20.9 | 18.2 | 17.7 | 15.9 | 17.7 | 17.1 | 17.4 |
| **Ours-RL-7B** | 39.1 | 41.8 | 28.2 | 36.4 | 39.5 | 44.9 | 32.6 | 39.0 | 38.2 |
| Δ | 14.6 | 10.9 | 10.0 | 11.8 | 2.9 | 7.2 | 4.3 | 4.8 | 6.8 |

rate, i.e., the proportion of suggestions that are deemed plausible, is reported as the final metric. As shown in Fig. 5, the SFT model demonstrates a clear advantage in plausibility, reflecting the benefits of careful supervised fine-tuning. Nonetheless, overall pass rates remain moderate, largely due to the inclusion of redundant examinations designed to verify elements of the differential diagnosis.

## 5.4 ABLATION STUDY

The ablation study investigates two key aspects: (1) the effectiveness of incorporating an rank-sensitive diagnosis reward with different $\tau$ values during RL (RL reward ablation), and (2) the contribution of the RAGES components (RAGES ablation).

**RL reward ablation.** We evaluate four variants of the diagnosis reward. (1) Binary reward: 0 or 1, where a hit receives 1 and otherwise 0. (2) Rank-sensitive reward with $\tau = 0.8$. (3) Rank-sensitive reward with $\tau = 1.5$. (4) Rank-sensitive reward with two $\tau$ values as the main setting: $\tau_1 = 0.8$ for cases with only primary information and $\tau_2 = 1.5$ for cases with additional evidence. We assess performance using three metrics: differential accuracy (DiffAcc), final diagnosis accuracy (DxAcc), and the average length of initial differential lists (#DDx) to directly show how the reward function shapes the length of the differential list.

Table 3: Ablation study on different diagnostic rewards.

| $R_d$ | | DiffAcc | DxAcc | #DDx |
|---|---|---|---|---|
| 0/1 | $\tau_1$ / $\tau_2$ | | | |
| ✓ | - | 62.4 | 31.4 | 9.56 |
| - | 0.8 / 0.8 | **65.2** | 32.5 | 8.66 |
| - | 1.5 / 1.5 | 57.2 | 31.6 | 7.25 |
| - | 0.8 / 1.5 | 64.8 | **38.2** | 7.39 |

Table 4: Ablation study on different phases of RAGES.

| RAGES | | Correctness (%) | | |
|---|---|---|---|---|
| w/ GT | w/ KB | EN | CN | Overall |
| | | 82.7 | 79.2 | 80.2 |
| ✓ | | **86.4** | 80.3 | 82.0 |
| | ✓ | 84.5 | 80.7 | 81.8 |
| ✓ | ✓ | **86.4** | **84.1** | **84.8** |

As shown in Table 3, a lower $\tau$ encourages longer differential lists (#DDx: 7.25 → 8.66 when changing $\tau$ from 1.5 to 0.8), improving DiffAcc. The binary 0/1 reward can be considered a limiting case of $\tau \to 0$, where all positions contribute equally, thus providing the longest list. A higher $\tau$ favors precision in the presence of sufficient information but underperforms at the early stage. Therefore, dynamically combining lower and higher $\tau$ values effectively guides the diagnostic process across cases with varying levels of available evidence.

**RAGES ablation.** Following Section 5.3, we employ GPT to evaluate the correctness of simulated examination results by DeepSeek-R1 under four RAGES configurations: vanilla generation, generation with reused ground-truth results (w/ GT), generation with retrieved knowledge (w/ KB), and the full combination of both reuse and retrieval. Results are reported in Table 4. Incorporating either reused or retrieved information improves output quality, while combining both strategies achieves the highest correctness. These findings confirm the complementary value of reuse and retrieval in producing examination results that are both factual and plausible.

# 6 CONCLUSION AND DISCUSSION

In this work, we propose an evidence-seeking two-turn reasoning paradigm for medical diagnosis, highlighting the importance of requesting and leveraging additional evidence. To realize this approach, we curate datasets of pathological cases, develop RAGES to simulate requested tests, and employ reinforcement learning with diagnostic evidence-seeking rewards to activate the reasoning process. Our experiments demonstrate that access to additional information substantially improves diagnostic accuracy. The models achieve competitive performance in pathological diagnosis, and RAGES generates more reliable and plausible examination results.

Despite these advances, several directions remain for future work. (1) While we provide an initial static theoretical analysis of $\tau$ and observe corresponding empirical effects, a deeper dynamic analysis of the reward mechanism would help better understand how the diagnostic incentives shape model behavior. (2) RAGES is currently used as an offline simulator due to computational constraints. Integrating RAGES directly into the RL training loop could enable multi-turn rollouts and further improve reasoning capabilities. (3) We primarily conduct experiments in pathological settings, as these naturally provide a clear stage separation between initial and follow-up diagnoses. While the core concept of evidence-seeking is broadly applicable, constructing datasets that span multiple medical specialties would enable a more systematic understanding of diagnostic reasoning. (4) This work primarily emphasizes diagnostic outcomes. Developing a more interpretable diagnostic process would require careful adjustment of the underlying reasoning steps, which could in turn motivate further research on process-based reward design and alignment with human clinician judgment.

Overall, this work presents a structured framework for interactive AI-assisted pathological diagnosis, laying the foundation for more evidence-driven, multi-turn reasoning in medical AI applications.

ETHICS STATEMENT

This work leverages pathological case studies from multiple sources, raising two primary ethical considerations, i.e., patient privacy and data re-distribution.

Regarding patient privacy, all case reports collected from public websites and journals were already anonymized at the source. We carefully removed any personally identifiable information from the in-house dataset, preserving only essential information such as age and gender for clinical reasoning.

Regarding data distribution, we will not release our in-house dataset publicly due to institutional data protection policies. For externally sourced cases, we strictly adhered to the usage guidelines specified by each website or journal. To avoid unauthorized redistribution, we will release only the URLs linking to the original case sources, allowing other researchers to access the materials while respecting the original data ownership and licensing terms.

The risk of this work may also lie in improper responses (including repetitive patterns, false information, and offensive output) since we do not specifically strengthen the safety of the model. The presented cases may contain content that is offensive.

## REPRODUCIBILITY STATEMENT

We will open-source the code of RL and URLs of public cases to facilitate reproduction. Meanwhile, all the prompts and the training settings are revealed in the Appendix, including the judging prompts, the RL and SFT dialogue prompts, the evaluation prompts, and the RAGES prompt.

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

## A  APPENDIX

### A.1  LLM USAGE STATEMENT

We used LLMs to refine the writing, including checking grammar, rephrasing, and improving readability. To ensure the writing quality, we further check and refine the generated text. Meanwhile, we used LLMs to find the prototype diagnostic reward function as $\frac{e^{-i}}{\sum_{k=1}^{|\mathbb{D}|} e^{-k}}$, which can be larger when the list is shorter or the hit position is earlier. And we develop it into the final form of the proposed rank-sensitive diagnostic reward.

### A.2  PROOF

**Theorem 1**  The rank-sensitive diagnosis reward confers a higher value when the correct diagnosis appears earlier in the diagnosis list. More formally, consider two diagnosis lists of equal length ($|\mathbb{D}^1| = |\mathbb{D}^2|$) that both contain the ground truth diagnosis $Y$. Let the position of the correct diagnosis in $\mathbb{D}^1$ be $i$ and in $\mathbb{D}^2$ be $j$, such that $D_i^1 = Y$ and $D_j^2 = Y$ with $1 \le i < j \le |\mathbb{D}^1| = |\mathbb{D}^2|$. In this case, $R_d(\mathbb{D}^1, Y) > R_d(\mathbb{D}^2, Y) > 0$.

*Proof.* According to the definition in Equation 3, we have,

$$R_d(\mathbb{D}^1, Y) = \frac{e^{-i/\tau}}{\sum_{k=1}^{|\mathbb{D}^1|} e^{-k/\tau}} = \frac{e^{-i/\tau}}{\sum_{k=1}^{|\mathbb{D}^2|} e^{-k/\tau}} > \frac{e^{-j/\tau}}{\sum_{k=1}^{|\mathbb{D}^2|} e^{-k/\tau}} = R_d(\mathbb{D}^2, Y)$$

$\square$

**Theorem 2**  The rank-sensitive diagnosis reward assigns a higher value when the diagnosis list is relatively short. More formally, consider two diagnosis lists $\mathbb{D}^1$ and $\mathbb{D}^2$ of different lengths ($|\mathbb{D}^1| < |\mathbb{D}^2|$) that both contain the ground truth diagnosis $Y$. Let the position of the correct diagnosis be $i$ in both lists, such that $D_i^1 = Y$ and $D_i^2 = Y$ with $1 \le i \le |\mathbb{D}^1| < |\mathbb{D}^2|$. In this case, $R_d(\mathbb{D}^1, Y) > R_d(\mathbb{D}^2, Y) > 0$.

*Proof.* According to the definition in Equation 3, we have,

$$R_d(\mathbb{D}^1, Y) = \frac{e^{-i/\tau}}{\sum_{k=1}^{|\mathbb{D}^1|} e^{-k/\tau}} > \frac{e^{-i/\tau}}{\sum_{k=1}^{|\mathbb{D}^1|} e^{-k/\tau} + \sum_{k=|\mathbb{D}^1+1|}^{|\mathbb{D}^2|} e^{-k/\tau}} = \frac{e^{-i/\tau}}{\sum_{k=1}^{|\mathbb{D}^2|} e^{-k/\tau}} = R_d(\mathbb{D}^2, Y)$$

$\square$

**Finding 1**  A lower $\tau$ in the rank-sensitive diagnosis reward encourages the correct diagnosis to the top of the diagnosis list. More formally, consider a fixed diagnosis list $\mathbb{D}$ that contains the ground truth diagnosis $Y$, where the position of $Y$ is 1 (i.e., the top-1 diagnosis). For two hyperparameters satisfying $\tau_1 < \tau_2$, we have $R_d^{\tau_1} \ge R_d^{\tau_2}$. The equality holds if and only if $|\mathbb{D}| = 1$.

*Proof.* Let $\tau_2 = \tau_1 + \Delta\tau$. After simplification, we have,

$$R_d^{\tau_1} - R_d^{\tau_2} = \frac{\sum_{k=2}^{|\mathbb{D}|} e^{-(\tau_1+k\tau_1)/\tau_1\tau_2}(e^{-\Delta\tau/\tau_1\tau_2} - e^{-k\Delta\tau/\tau_1\tau_2})}{\sum_{j=1}^{|\mathbb{D}|} e^{-j/\tau_1} \sum_{j=1}^{|\mathbb{D}|} e^{-j/\tau_2}}$$

.

When $|\mathbb{D}| > 1$, we have $R_d^{\tau_1} - R_d^{\tau_2} > 0$, and when $|\mathbb{D}| = 1$, $R_d^{\tau_1} = R_d^{\tau_2} = 1$. $\square$

**Finding 2**  A higher $\tau$ encourages the generation of shorter diagnosis lists. More formally, consider two diagnosis lists of differing lengths, where $|\mathbb{D}^1| < |\mathbb{D}^2|$, and let the correct diagnosis $Y$ appear at position $i$ in both lists. Define the reward increment with respect to $\tau$ as $\Delta R_d^\tau = R_d^\tau(\mathbb{D}^1, Y) - R_d^\tau(\mathbb{D}^2, Y)$. For hyperparameters satisfying $\tau_1 > \tau_2$, we have $\Delta R_d^{\tau_1} > \Delta R_d^{\tau_2} > 0$.

*Proof.* We have $\Delta R_d^\tau = R_d(\mathbb{D}^1, Y) - R_d(\mathbb{D}^2, Y) = \frac{e^{-i/\tau}}{S_{1,a}S_{1,b}}S_{a+1,b} > 0$ (proved in Theorem 2), where $S_{a,b} = \sum_{k=a}^{b} e^{-k/\tau}$, $a = |\mathbb{D}^1|$ and $b = |\mathbb{D}^2|$.

Its log-derivation is $(\log \Delta R_d^\tau)' = \frac{i}{\tau^2} + \frac{1}{\tau^2}\left(T_{a+1,b} - T_{1,a} - T_{1,b}\right)$, where $T_{a,b} = \frac{\sum_{k=a}^{b} ke^{-k/\tau}}{S_{a,b}}$.

With several simplifications, we have $F = (1 - T_{1,a})S_{1,b} + (T_{a+1,b} - T_{1,a})S_{1,a}$, which has the same sign with $(\log \Delta R_d^\tau)'$.

It is clear that $S_{1,a}$ is a geometric sequence with the initial item $e^{-1/\tau}$ and a ratio $q = e^{-1/\tau}$. By using the summation formula, we have $S_{1,a} = \frac{q(1-q^a)}{1-q}$.

Then we have

$$
\begin{aligned}
F_2 &= \frac{F}{q/(1-q)} \\
&= (1 - T_{1,a})(1 - q^b) + (T_{a+1,b} - T_{1,a})(1 - q^a) \\
&= (1 - T_{1,a} + T_{a+1,b} - T_{1,a}) - q^b(1 - T_{1,a}) - q^a(T_{a+1,b} - T_{1,a}) \\
&> (1 - T_{1,a} + T_{a+1,b} - T_{1,a}) - q^a(1 - T_{1,a}) - q^a(T_{a+1,b} - T_{1,a}) \\
&= (1 + T_{a+1,b} - 2T_{1,a})(1 - q^a)
\end{aligned}
$$

$T_{a,b}$ can be seen as a weighted sum over $\{k | a \leq k \leq b\}$ with monotonically decreasing weights $\{e^{-k/\tau} | a \leq k \leq b\}$. Therefore, $T_{a,b}$ should be larger than the lower bound $a$ while less than its arithmetic mean $\frac{a+b}{2}$.

Hence we have $T_{a+1,b} > a + 1 = 2\frac{a+1}{2} > 2T_{1,a}$. By using all things ahead, we have,

$$
\begin{aligned}
(\log \Delta R_d^\tau)' \propto F \propto F_2 \\
> (1 + T_{a+1,b} - 2T_{1,a})(1 - q^a) \\
> 1 - q^a > 0
\end{aligned}
$$

Therefore, $\Delta R_d^\tau$ is strictly monotonically increasing with $\tau$, and thus we have $\Delta R_d^{\tau_1} > \Delta R_d^{\tau_2} > 0$ when $\tau_1 > \tau_2$.

$\square$

Table 5: Performance of models under different settings: a direct diagnosis fashion and the proposed evidence-seeking paradigm. **Bold** figures suggest the best performance, and the underlined are the second best. For cells that contain two numbers, the first represents the performance, and the subscript indicates the change compared with the direct diagnosis setting (positive values in green, negative in red). For clarity and consistency, all subscripts are rounded to two significant digits.

| Model | Direct DxAcc | | | Evidence-seeking DxAcc | | | Evidence-seeking DiffAcc | | |
| | EN | CN | Mean | EN | CN | Mean | EN | CN | Mean |
| --- | --- | --- | --- | --- | --- | --- | --- | --- | --- |
| Qwen2.5-32B | 33.9 | 31.7 | 32.3 | $24.5_{9.4}$ | $34.2_{2.5}$ | $31.4_{0.9}$ | $42.7_{8.8}$ | $51.9_{20.}$ | $49.3_{17.}$ |
| Qwen2.5-72B | **43.3** | 37.3 | 39.0 | $\mathbf{43.9}_{0.6}$ | $45.4_{8.0}$ | $45.0_{6.0}$ | $54.2_{11.}$ | $65.6_{28.}$ | $62.4_{23.}$ |
| QwQ-32B | 37.3 | $\underline{43.7}$ | 41.9 | $42.7_{5.4}$ | $50.4_{6.7}$ | $48.2_{6.3}$ | $57.6_{20.}$ | $66.1_{22.}$ | $63.6_{22.}$ |
| Qwen3-32B | 36.0 | 42.7 | 40.8 | $29.7_{6.3}$ | $43.1_{0.4}$ | $39.3_{1.5}$ | $44.3_{8.3}$ | $56.3_{14.}$ | $\underline{52.8}_{12.}$ |
| Huatuo-7B | 22.4 | 24.0 | 23.6 | $19.7_{2.7}$ | $21.1_{2.9}$ | $20.7_{2.9}$ | $31.5_{9.1}$ | $40.1_{16.}$ | $37.6_{14.}$ |
| M2-32B | 37.6 | **44.3** | $\underline{42.4}$ | $37.9_{0.3}$ | $49.2_{4.9}$ | $46.0_{3.6}$ | $52.7_{15.}$ | $66.0_{22.}$ | $62.2_{20.}$ |
| Ours-SFT-32B | $\underline{40.9}$ | 43.6 | **42.8** | $42.7_{1.8}$ | $\mathbf{51.3}_{7.7}$ | $\mathbf{48.9}_{6.1}$ | $54.9_{14.}$ | $\mathbf{67.0}_{23.}$ | $63.6_{21.}$ |
| Ours-RL-7B | 29.1 | 34.4 | 32.9 | $\underline{36.4}_{7.3}$ | $39.0_{4.6}$ | $38.2_{5.3}$ | $\mathbf{62.0}_{33.}$ | $65.8_{31.}$ | $\mathbf{64.8}_{32.}$ |

A.3 IDEALIZED EXPERIMENT: ONE-TURN DIAGNOSIS WITH MORE EVIDENCE

To better mitigate the information discrepancy, we conduct an idealized one-turn diagnosis experiment in which models are provided with as much clinical information as possible. This setting is intentionally "ideal," as real-world clinical diagnosis does not permit access to future follow-up results. The outcomes are summarized in Table 5.

Overall, evidence-seeking remains beneficial for most models. However, Qwen2.5-32B and Qwen3-32B achieve higher accuracy when directly prompted to provide diagnoses for the English cases. A plausible explanation is that these English cases contain sufficient examination findings (whereas some Chinese cases may lack such details), which are better suited to these baseline models, not tailored to an evidence-seeking paradigm. Huatuo-7B, which shows consistently degraded performance, may suffer from strong adherence to its training patterns, i.e., the traditional one-turn diagnosis. This issue is also observed in the main experiments, where it tends not to follow the instructed output format but defaults to its own pattern.

In contrast, the differential accuracy provides a more encouraging signal. It can be viewed as an approximate upper bound for the evidence-seeking paradigm, suggesting substantial room for improvement: not only by enriching the available information but also by enhancing models' ability to integrate and leverage appended clinical evidence effectively.

A.4 PERFORMANCE ON GENERAL MEDICAL TASKS: MEDQA

To evaluate whether task-specific training influences general performance on medical tasks, we report the results of our models alongside their corresponding base models on the MedQA dataset (Jin et al., 2021), including both the English and Simplified Chinese test subsets (Table 6).

For the RL-trained model, performance differences are minimal, suggesting that its original medical capabilities are largely retained. In contrast, the SFT-trained model exhibits larger differences. Nevertheless, likely due to the narrow focus on a single diagnostic task and the limited number of cases, we do not observe a general improvement in zero-shot medical QA performance.

Table 6: Performance of models on MedQA zero-shot QA task. For cells that contain two numbers, the first represents the performance, and the subscript indicates the change compared with the base model (positive values in green, negative in red). For clarity and consistency, all subscripts are rounded to two significant digits.

| Model | English (1273) | Simplified Chinese (3426) |
|---|---|---|
| Qwen2.5-7B | 53.5 | 82.8 |
| Qwen2.5-32B | 68.5 | 89.7 |
| Ours-RL-7B | $52.5_{1.0}$ | $82.9_{0.1}$ |
| Ours-SFT-32B | $74.0_{5.5}$ | $87.1_{2.6}$ |

A.5 PERFORMANCE OF *Ours-SFT-7B*

To further investigate the performance differences between SFT and RL models, we trained an SFT model of the same size (7B) as the RL model. Table 7 reports the performance of Huatuo-7B (as a baseline), Ours-SFT-7B, and Ours-RL-7B on both English (EN) and Chinese (CN) test sets, in terms of differential diagnosis accuracy (DiffAcc) and final diagnosis accuracy (DxAcc). From the results, several observations can be made:

Performance gains from task-specific training. Both Ours-SFT-7B and Ours-RL-7B substantially outperform the Huatuo-7B baseline trained with general medical data across all metrics. Moreover, RL training yields the most significant improvements. Compared with SFT, RL training produces notable additional gains in both DiffAcc and DxAcc. Specifically, Ours-RL-7B improves the mean DxAcc from 30.1 (SFT) to 38.2, indicating that reinforcement learning better aligns the model with the rank-sensitive reward and evidence-seeking behavior, resulting in more accurate final diagnoses.

Table 7: Performance comparison of Huatuo-7B, Ours-SFT-7B and Ours-RL-7B.

| Model | DiffAcc | | | DxAcc | | |
|---|---|---|---|---|---|---|
| | EN | CN | Mean | EN | CN | Mean |
| Huatuo-7B | 31.5 | 40.1 | 37.7 | 19.7 | 22.5 | 21.7 |
| Ours-SFT-7B | 45.5 | 51.2 | 49.6 | 29.4 | 30.4 | 30.1 |
| Ours-RL-7B | 62.0 | 65.8 | 64.8 | 36.4 | 39.0 | 38.2 |

## A.6 DISCUSSION ON EFFICIENCY

In the main text, we report that the training process takes approximately 40 hours on 8 H100 GPUs. Time consumption arises from two main sources: the basic RL training (rollout generation, gradient computation, etc.) and the computationally intensive model-based judging process (diagnosis and examination assessment). We provide further details below.

**Inefficient GPU usage with LLM judges.** Using collocation and sleep strategies, OpenRLHF efficiently allocates GPUs across models with different roles. However, we could not identify a feasible interface to integrate an additional model for judgment that directly returns text. Consequently, we adopted a less efficient approach: we dedicated 4 GPUs exclusively to the judge model and relied on HTML to transfer information between models.

**LLM judges are computationally expensive.** Beyond the framework overhead, the judge models themselves incur significant cost. For each generation, the judge model is used twice: once for diagnostic evaluation to promote accurate diagnoses, and once for examination assessment to ensure reasonable test requests, effectively doubling the computational load. In a trial experiment without the examination judge, training time dropped to roughly 28 hours. Using a smaller judge, such as Qwen2.5-7B, nearly halved the judging cost, but led to severe counting errors and incorrect judgments. Therefore, we ultimately hosted a Qwen2.5-32B model on 4 GPUs for the judging role, despite the inefficiency.

Table 8: Inference efficiency of models. Hosted on 4 H100 GPUs with vLLM engine.

| Model | Initial | | Follow-up | |
|---|---|---|---|---|
| | Inference Speed (it/s) | #Token | Inference Speed (it/s) | #Token |
| Qwen2.5-32B | 6.02 | 414.1 | 4.55 | 719.3 |
| Qwen2.5-72B | 3.18 | 605.6 | 3.8 | 858.8 |
| QwQ-32B | 1.71 | 2064.9 | 2.91 | 1114.8 |
| Qwen3-32B | 4.21 | 1455.5 | 5.38 | 824.1 |
| Huatuo-7B | 10.38 | 594.7 | 22.1 | 401.1 |
| M2-32B | 1.02 | 1160.1 | 5.11 | 381.6 |
| Ours-SFT-32B | 1.27 | 2244.7 | 2.02 | 2462.4 |
| Ours-RL-7B | 3.37 | 1154.9 | 4.76 | 1002.7 |

For inference efficiency, we report the average inference speed (items per second, tested on 4 H100 GPUs with vLLM engines) and the average number of generated tokens per model in Table 8.

From the results, several observations can be made:

**Reasoning models typically generate more tokens than chat models.** For instance, Ours-SFT-32B produces 2244.7 tokens in the initial turn compared with 414.1 tokens for Qwen2.5-32B. This increase in token generation naturally slows down the inference speed, as seen in Ours-SFT-32B's lower speed (1.27 it/s) relative to chat models like Ours-SFT-32B (6.02 it/s).

**Follow-up turns often generate fewer tokens in some models, but our models show the opposite tendency.** For example, Qwen3-32B generates 1455.5 tokens in the initial turn at 4.21 it/s and only 824.1 tokens in the follow-up turn at 5.38 it/s. In contrast, Ours-SFT-32B produces more tokens in the follow-up (2462.4) than in the initial turn (2244.7), which slightly slows down its speed (2.02 it/s). This may suggest the verbosity of our models in the follow-up stage.

**Taking an additional turn generally increases total computational consumption.** The follow-up turn incurs the least overhead with M2-32B, achieving roughly 5x speed and generating 32.8% more tokens. Nevertheless, this additional cost remains non-negligible.

## A.7 PROMPTS IN USE

Here is the prompt used in RL training.

---

**RL Training Template**

```
system:
You are Qwen, created by Alibaba Cloud. You are a helpful
    assistant. A conversation between User and Assistant. The user
    asks a question, and the Assistant solves it. The assistant
    first drafts the reasoning process (inner monologue) until it
    has derived the final answer with full confidence. It then
    provides a self-contained summary of the thoughts, i.e.,
    keeping succinct but containing all the critical steps needed
    to reach the conclusion. It should use Markdown and Latex to
    format the response. Write both the thoughts and summary in the
    same language as the task posed by the user.\n\n The thinking
    process must follow the template below (You should **include
    and only include one** pair of <think></think> and
    <answer></answer> tags in your response): \n<think>\n The
    thoughts or/and draft, like working through an exercise on
    scratch paper. Be as casual and as long as necessary until it
    is confident to generate a correct
    answer.\n</think>\n\n<answer>\n Here, provide a concise summary
    that reflects the reasoning process and presents a clear final
    answer to the user.\n</answer>\n

user:
I need you to infer possible differential diagnoses and related
    additional tests based on the provided case information.
    Specifically, you need to analyze the given case information
    carefully. Then, based on the evidence points within it,
    gradually deduce all possible related differential diagnoses
    (these diagnoses must be relatively specific, avoiding simple
    descriptions such as "benign/malignant lesion"). Then,
    re-examine the case information to exclude those differential
    options with extremely low probability (actively reflect: does
    the existing information contain anything that clearly
    conflicts with this disease?). At the same time, you may also
    add new differential diagnoses (actively reflect: have I missed
    any possibilities?). You can engage in such reflection even
    during the initial inference. After repeating this process for
    multiple rounds, when you are sufficiently confident that the
    current set of differential diagnoses has a high probability of
    including the final diagnosis while having an extremely low
    likelihood of including unrelated diagnoses, you may proceed to
    the final summary stage. When summarizing the final
    differential diagnoses, rank those with a higher probability
    first. Note that these differential diagnoses should not exceed
    ten. Then, based on these differential diagnoses, provide the
    additional information needed to rule out some of the possible
    diagnoses (such as a specific immunohistochemical antibody, a
    specialized histochemical stain, or a particular molecular
    test). While thinking, you may review previous content at any
    time for reflection and promptly revise your conclusions as
    needed.
```

```
After thinking, you need to organize your response: briefly
    summarize your thought process, then summarize your output in
    the specified format.
Format requirements are as follows:
Differential Diagnoses: \DiffList{Diagnosis 1, Diagnosis 2, ...}
Further Examination Items: \ExamList{Item 1, Item 2, ...}
The following is the case information:
<Case Information>
```

Here is the full judge prompt for the proposed diagnosis.

**Diagnosis Judging Prompt**

```
I need you to act as a professional pathologist. After
    understanding what diseases the given candidate diagnoses and
    the actual diagnosis each represent (note: you only need to
    focus on the primary diagnostic conclusion in the actual
    diagnosis), analyze one by one whether each candidate diagnosis
    is the same as the actual diagnosis. To help you make a better
    judgment, I will also provide you with the differential
    diagnoses related to the actual diagnosis (the actual diagnosis
    is included among them). When a candidate diagnosis aligns with
    most of the differential diagnoses, you cannot consider that
    candidate diagnosis to be consistent with the actual diagnosis.
    You also need to record the corresponding position.
If the candidate diagnoses contain a diagnosis that is clearly
    contradictory to the actual diagnosis (e.g., the candidates
    include testicular cancer but the true diagnosis is ovarian
    cancer), or if a candidate largely aligns with most of the
    differential diagnoses, or if it refers to a disease that does
    not actually exist (e.g., "Aba Aba lymphoma"), the position is
    recorded as -1. Next, if there is no identical candidate
    diagnosis, but the candidate is related to the actual diagnosis
    in some way (easily confused / anatomically close / similar
    manifestation / easy to associate), then the position is
    recorded as 0. If there is an identical diagnosis among the
    candidates, calculate the position of the matching diagnosis
    among all candidate diagnoses (the position of the first one is
    1, followed by 2, 3, ...). You also need to record the total
    number of all candidate diagnoses.
Output format requirements:
After analyzing all candidate diagnoses, separate your analysis
    from the final result with ---. Then, if there is an identical
    diagnosis, output exactly one line: \boxed{True}+\pos{position
    / total}; if there is none, output \boxed{False}+\pos{0 /
    total}; if there is a clearly contradictory diagnosis, output
    \boxed{False}+\pos{-1 / total}; if the input is None, output
    \boxed{False}+\pos{0 / 0}.
Ground truth diagnosis: {...}
Reference differential diagnoses: {...}
Candidate diagnoses: {...}
```

Here is the full judge prompt for proposed examinations.

**Examination Judging Prompt**

```
I need you to act as a professional pathologist to help me
    determine whether the provided examination items can assist in
    differentiating among the given differential diagnoses. At the
    same time, I need you to assess whether the items are valid
```

```
        (i.e., whether any content has been fabricated, such as CD1355;
        or whether any items are overly broad, such as simply stating
        "molecular testing" without specifying which particular test).
        Additionally, these examination items should be of the type
        typically used in pathology, such as specific
        immunohistochemical antibodies, special histochemical stains,
        or particular molecular tests. They should not include tests
        from other departments (such as imaging studies or blood tests).
If the provided examination items can effectively differentiate
        among the given diagnoses and contain no erroneous content,
        output 1. If the provided examination items cannot
        differentiate among the given diagnoses, but contain no
        erroneous content, output 0. If the provided examination items
        contain fabricated content or involve tests from other
        departments, output -1. If the given diagnoses contain obvious
        errors or are too broad, making it impossible to propose
        reasonable additional tests, also output -1.
Output format requirement:\boxed{{1|0|-1}}
Examination items: {...}
Differential diagnoses: {...}
```

Here is the RAGES prompt.

**RAGES Prompt**

```
Based on the given information, after careful consideration, infer
        the possible result of each examination item. The given
        information includes the final diagnosis, examination items,
        existing results (if any), and relevant knowledge (if any).
        Specifically, you need to:

0. Only focus on the content that can produce definitive results.
1. First, check the "Existing Results" and record results that
        overlap with the examination. The confidence level is 1.
2. Then, check the "Relevant Knowledge". First, determine whether
        the relevant knowledge pertains to the same disease as
        described in the "Final Diagnosis". If it is the same disease,
        then, based on this knowledge, infer the results of the
        remaining examination items. The confidence level is 0.8.
3. Retrieve your own knowledge and speculate on the results of the
        remaining items. The confidence level is 0.6.
4. Output the above results in the specified format. The format is
        as follows:
        ExamRes: {"Item 1": ("Result 1", Confidence Level 1), "Item 2":
        ("Result 2", Confidence Level 2)}

Information provided:
Final Diagnosis: {...}
Examinations: {...}
Existing Results: {...}
Relevant Knowledge: {...}
```

Here is the SFT training prompt.

**SFT Prompt Template**

```
system:
You are Qwen, created by Alibaba Cloud. You are a helpful
        assistant. A conversation between User and Assistant. The user
        asks a question, and the Assistant solves it. The assistant
        first thinks about the reasoning process in the mind and then
```

```
1134
1135            provides the user with the answer. The reasoning process and
1136            answer are enclosed within <think> </think> and <answer>
1137            </answer> tags, respectively, i.e., <think> reasoning process
1138            here </think> <answer> answer here </answer>.
1139
1140     user:
1141     I need you to act as a professional pathologist. After carefully
1142            considering the given information, infer the possible
1143            differential diagnoses. Then, based on these differential
1144            diagnoses, suggest additional information that needs to be
1145            provided to rule out certain possibilities. Specifically:
         1. First, you need to carefully analyze the given information,
1146            which mainly includes case background information, previous
1147            examination items, morphological descriptions of pathological
1148            sections, etc. Summarize the evidence points related to the
1149            diagnosis from this information.
         2. Based on the given information, analyze what the possible
1150            differential diagnoses are and determine whether they are
1151            consistent with the given information. Note: These differential
1152            diagnoses should be as broad and accurate as possible (broad
1153            means considering less common diagnostic possibilities, and
1154            accurate means the listed differential diagnoses should not
1155            conflict with most of the background information).
         3. According to the listed differential diagnoses, propose the
1156            further examination items. You need to specify the exact
1157            antigen – antibody, staining type, or molecular type. If the
1158            existing information is sufficient to confirm a specific
1159            disease, only output that disease and leave the additional
1160            examination items blank.
         4. Finally, summarize the possible differential diagnoses and the
1161            required additional examination items in a given format. When
1162            summarizing the differential diagnoses, you need to rank the
1163            more likely diagnoses higher.
         5. Output four sections in the specified format: "Case Analysis –
1164            Differential Diagnosis Analysis – Additional Examination Items
1165            – Summary".
1166
         Format requirements:
1167     ## Case Analysis
1168     ...
1169     ## Summary
1170     **Differential Diagnoses**: \DiffList{Differential Diagnosis 1,
1171            ...}
1172     **Further Examinations**: \ExamList{Examination Item 1, ...}
1173
         Here is the case:
1174     <Case Information>
1175
1176
1177     assistant:
1178     ...
1179
1180     user:
1181     Now the results of the further examinations have come out. I need
            you to:
1182     1. First, check the "Case Information" and the "First-round
1183            Diagnosis" to sort out the previous diagnostic chain of thought
1184            and related conclusions.
         2. Then, check the "Results of Further Examinations". The
1185            additional test results may not fully match the items requested
1186            in the initial diagnosis. Based on the available test results,
1187            you need to conduct further differential analysis, and give the
```

```
     final diagnosis. Note: You are completely entitled to overturn
     the initial diagnostic approach and provide a diagnosis based
     on the current information after obtaining more data.
3. The final diagnosis must be output in the specified format,
     i.e., \boxed{Diagnosis Name}
Here is the information:
Results of Further Examinations: <Exam Results>

assistant:
...
```

Here are the evaluation prompts for diagnoses and examinations.

**Evaluation Prompts**

```
user:
I need you to act as a professional pathologist. After careful
     consideration based on the given disease candidates and the
     true diagnosis, determine whether the true diagnosis (or a
     close approximation) is among the candidates and, if present,
     its position in the list. If it is within the candidates,
     output \boxed{True} + "Hit candidate content" + "Position of
     the hit content" at the end; otherwise, output \boxed{False} +
     No hit + 0.

Ground truth diagnosis: {...}
Candidate diagnoses: {...}

assistant:
...
\boxed{True | False} + ... + <digital>

user:
I need you to assist me in determining whether some pathological
     content is reasonable. I will provide you with a list of
     differential diagnosis diseases, a set of further examination
     results, and the ground truth diagnosis. You need to determine:
1. Based on the list of differential diagnosis diseases, judge
     whether the additional examination items are reasonable and
     record the unreasonable items;
2. Based on the ground truth diagnosis, judge whether the further
     examination results are reasonable and record the incorrect
     results.

**Notes**:
1. When the additional examination items are "no need," both items
     can be directly considered reasonable.
2. When judging the plausibility of examination results, do not
     consider whether some results are omitted; only judge the
     reasonableness of the existing examination results.

The information you need to use is as follows:
- Differential diagnosis: {...}
- Further examinations and results: {...}
- Ground truth diagnosis: {...}

After careful consideration, you need to summarize at the end of
     the output in the following format:
```

```
1. Exam: \boxed{True|False}, \List{Wrong Item 1, ...}
2. Result: \boxed{True|False}, \List{Wrong Item and Result 1, ...}

assistant:
...
1. Exam: \boxed{True | False}, \List {...}
2. Result: \boxed{True | False}, \List {...}
```

## A.8 SUPERVISED FINE-TUNING DETAILS

Following Huang et al. (2025), we create an SFT dataset generated by DeepSeek-R1 to activate the reasoning of existing LLMs on two-turn diagnosis.

In the **initial consultation**, LLMs take clinical history and histological findings as input and think about possible differential diagnoses and appended examinations. Instead of directly giving the differential list in the original cases as a guide, we adopt a post-verification strategy. We only provide the input and generate it several times for each case. After gathering these trials, we use another LLM to judge whether the ground truth diagnosis appears in these trials, and retain the positive ones as SFT data. We chose this more complex strategy for two reasons. First, we do not expect LLMs to be bound by the original text, since different pathologists might derive different disease suspects from the same case. Second, when a differential list is provided, LLMs might have hallucinations like direct references to the original results.

Based on the verified trials, we use the RAGES method to simulate the appended results. After collecting sufficient appended results, we can start the **follow-up consultation**. We provide LLMs with the response in the first turn and the acquired further examination results, and ask them to produce a final diagnosis after careful thinking. Also, we employ the post-verification strategy and retain as SFT data those trajectories that propose the true diagnosis.

A total of 925 training samples related to initial consultations and 623 follow-up samples are constructed. For SFT, we adopt Qwen2.5-32B-Instruct (Yang et al., 2024) as the base model. We apply parameter-efficient Low-Rank Adaptation (LoRA) (Hu et al., 2022) and enable bf16 precision to optimize training with our curated dataset. The training workflow is implemented using Llama-Factory (Zheng et al., 2024), and evaluation is conducted with vLLM (Kwon et al., 2023). We use LoRA with default hyperparameters, as $\alpha = 16$ and $r = 8$. The initial learning rate is set to $5 \times 10^{-5}$ with cosine decay, and training is run for 15 epochs. The fine-tuning is carried out on 8 A100 GPUs, and the entire process completes in approximately 12 hours.

## A.9 DATASET DETAILS

The training dataset comes from two sources:

- **DakaPath** is a Chinese platform for pathological teaching and communication. In addition to its extensive knowledge, DakaPath features a special section called Micro Lecture, which provides expert explanations of hundreds of real-life cases. We collected 373 raw explanations.

- **Chinese Journal of Pathology** reports on advanced scientific research achievements and pathological diagnosis experience as case discussions. Initially, we included 653 cases.

The evaluation dataset consists of cases from:

- **Pathology Outlines** is a comprehensive platform for pathology communication. It offers extensive knowledge across subspecialties and weekly case studies. We collected 483 cases published before December 2024.

- **Hans Popper Hepatopathology Society (HPHS)** is a hepatopathology-focused community that presents a noteworthy case every 3 to 4 months. We included 37 cases published up to September 2024.

- The **in-house dataset** originates from internal examination materials used for resident education at Hospital X. It includes 339 cases covering 11 major domains, including the endocrine system, reproductive system, respiratory system, central nervous system, skin, bone and soft tissue, etc.

## A.10 OTHER DETAILS

Here are the detailed hyperparameters used in the RL algorithm. If not specified, a default value is adopted.

Table 9: Hyperparameters used in RL.

| Hyperparameter | Value | Hyperparameter | Value |
|---|---|---|---|
| train batch size | 16 | actor learning rate | 5e-7 |
| micro train batch size | 2 | init kl coef | 0.001 |
| rollout batch size | 64 | prompt max len | 4096 |
| micro rollout batch size | 4 | generate max len | 4096 |
| n samples per prompt | 4 | num episodes | 30 |

## A.11 CASE STUDY

We first present a case generated with our RL model to demonstrate the whole process.

---

**Case 1: Hepatocellular carcinoma**

**Case Information:** A 65 year old man presented with a large mass in the floor of the mouth (levels Ia to VI) but mostly suprahyoid. Ultrasound showed a vascular appearance and a normal appearing thyroid gland. The patient also had a prior history of hepatocellular carcinoma, treated with TACE, with a rising serum alpha fetoprotein.

The core biopsy shows a trabecular pattern of tumor cells separated by tortuous sinusoid-like vascular channels. At high power, the pleomorphic tumor cells have well defined cytoplasmic boundaries, a hepatoid appearance with abundant pink cytoplasm, and prominent nucleoli. Bile plugs are easily seen.

**Ground Truth Diagnosis:** Hepatocellular carcinoma

**Initial Prediction:** First, based on the case information, the patient is a 65-year-old male whose main symptom is a large mass on the floor of the mouth, extending from level Ia to VI, but primarily located suprahyoid. Ultrasound shows a vascular appearance of the mass, with a normal thyroid gland. The patient has a history of hepatocellular carcinoma, has undergone TACE treatment, and has an elevated serum alpha-fetoprotein (AFP).

From the histological description, the tumor cells are arranged in a trabecular pattern, with tortuous sinusoid-like vascular structures. Under high magnification, the tumor cells are pleomorphic, with well-defined cytoplasmic boundaries, a hepatoid appearance, abundant pink cytoplasm, and prominent nucleoli. In addition, bile plugs are present.

Based on this information, we can consider the following possible differential diagnoses: Hepatocellular carcinoma, Cholangiocarcinoma, Metastatic cancer, Lymphoma, Acinar cell carcinoma, Metastatic hepatocellular carcinoma, Metastatic cholangiocarcinoma, Metastatic lymphoma, Metastatic acinar cell carcinoma, Metastatic neuroendocrine tumor.

Next, we re-examine these diagnoses and exclude options with extremely low probability. By analyzing the case information, we can see that the mass is located on the floor of the mouth, the patient has a history of hepatocellular carcinoma, and the serum AFP is elevated. These clues suggest that hepatocellular carcinoma and cholangiocarcinoma are relatively high-probability diagnoses. Metastatic cancer, lymphoma, and acinar cell carcinoma are also possible diagnoses. The possibility of neuroendocrine tumor is relatively low, as the case information does not mention neuroendocrine features.

---

After reflection, we find that hepatocellular carcinoma and cholangiocarcinoma are the most likely diagnoses, because the patient has a history of hepatocellular carcinoma and the mass exhibits features of both hepatocytes and bile duct cells. Metastatic cancer, lymphoma, and acinar cell carcinoma are also possible diagnoses, but with relatively lower probability. The possibility of neuroendocrine tumor is relatively low, as the case information does not mention neuroendocrine features.

In conclusion, the differential diagnoses we consider include: Hepatocellular carcinoma, Cholangiocarcinoma, Metastatic cancer, Lymphoma, Acinar cell carcinoma, Metastatic hepatocellular carcinoma, Metastatic cholangiocarcinoma, Metastatic lymphoma, Metastatic acinar cell carcinoma.

Differential Diagnoses: \DiffList{Hepatocellular carcinoma, Cholangiocarcinoma, Metastatic cancer, Lymphoma, Acinar cell carcinoma, Metastatic hepatocellular carcinoma, Metastatic cholangiocarcinoma, Metastatic lymphoma, Metastatic acinar cell carcinoma}

Additional Examination Items: \ExamList{Immunohistochemistry (e.g., AFP, CK, Vimentin, CD3, CD20, ACCP, S100, CK19, CK7, CD11c, CD68, CD117, CD34, Ki-67, etc.), Special histochemical stains (e.g., PAS, AB-PAS, etc.), Molecular testing (e.g., gene expression analysis for RAS, KRAS, BCL2, BCL6, MALT1, etc.)}

**Follow-up Prediction:**

First, we review the initial round of diagnostics. The first round considered several possibilities: hepatocellular carcinoma, cholangiocarcinoma, metastatic carcinoma, lymphoma, acinar epithelial cell carcinoma, metastatic hepatocellular carcinoma, metastatic cholangiocarcinoma, metastatic lymphoma, metastatic acinar epithelial cell carcinoma. Based on the case information, the mass is located on the floor of the mouth, with a history of hepatocellular carcinoma and an elevated serum alpha-fetoprotein (AFP). Histological description indicates that the tumor cells are arranged in a trabecular pattern, with tortuous sinusoid-like vascular structures. Under high magnification, the tumor cells appear pleomorphic, with well-defined cytoplasmic boundaries, a hepatoid appearance, abundant pink cytoplasm, and prominent nucleoli. Additionally, bile plugs are present.

Next, we examine the results of the additional tests: AFP: Positive CK: Positive Vimentin: Negative CD3: Negative CD20: Negative ACCP: Positive S100: Negative CK19: Negative CK7: Negative CD11c: Negative CD68: Negative CD117: Negative CD34: Positive Ki-67: 20% Positive RAS: Wild-type KRAS: Wild-type BCL2: Negative BCL6: Negative MALT1: Negative

Based on these results, we can further analyze:

Both AFP and ACCP are positive, suggesting possible hepatocellular carcinoma or cholangiocarcinoma.

Vimentin, CD3, and CD20 are all negative, suggesting the tumor is not of mesenchymal or lymphoid origin.

CD34 is positive, suggesting a possible epithelial origin.

Ki-67 is 20% positive, indicating moderate tumor cell proliferation activity.

Both RAS and KRAS are wild-type, suggesting the tumor does not harbor certain specific gene mutations.

BCL2, BCL6, and MALT1 are all negative, suggesting it is not lymphoma or MALT lymphoma.

Synthesizing this information, we can narrow the diagnostic possibilities to:

Hepatocellular carcinoma Cholangiocarcinoma Acinar epithelial cell carcinoma Metastatic hepatocellular carcinoma Metastatic cholangiocarcinoma Metastatic acinar epithelial cell carcinoma

Through analysis, we find that the positivity of both AFP and ACCP suggests hepatocellular carcinoma or cholangiocarcinoma. CD34 positivity suggests an epithelial origin. The 20% Ki-67 positivity indicates moderate proliferative activity. Wild-type RAS and KRAS suggest the absence of specific mutations. The negativity of BCL2, BCL6, and MALT1 rules out lymphoma or MALT lymphoma.

In conclusion, our final diagnostic considerations include: Hepatocellular carcinoma Cholangiocarcinoma Acinar epithelial cell carcinoma Metastatic hepatocellular carcinoma Metastatic cholangiocarcinoma Metastatic acinar epithelial cell carcinoma

Final Diagnosis: \boxed{Hepatocellular carcinoma}

We then present a case generated with our RL model, which produces a vague diagnosis in the initial turn and refines it into a precise one in the second turn.

---

**Case 2: Diagnosis Refinement**

**Case Information:** An HIV seronegative, 96 year old African-American woman with no remarkable past medical history presented with a left upper eyelid lesion, which was completely excised. The specimen consisted of skin and mucosa-covered soft tissue. Immediately subjacent to the hairy edge was a 0.7 x 0.5 x 0.5 cm indurated nodule, with a mottled brown tan cut surface. Histological assessment showed a circumscribed stromal tumor, with spindle cells forming vascular spaces, extravasated red blood cells and occasional hyaline globules.

**Ground Truth Diagnosis:** Kaposi sarcoma

**Initial Prediction:** First, based on the case information, the patient is a 96-year-old African American female with no significant past medical history, presenting with a lesion on the left upper eyelid. The pathological findings reveal a well-circumscribed stromal tumor composed of spindle cells forming vascular spaces, accompanied by red blood cell extravasation and occasional hyaline bodies.

Initial Differential Diagnoses: Fibroma, Angiofibroma, Metastatic tumor, Metastatic melanoma, Metastatic neuroendocrine tumor, Metastatic lymphoma, Metastatic lipoma, Metastatic leiomyoma, Metastatic epithelial tumor, Metastatic mesenchymal tumor

Reflection and Adjustment
Fibroma: Although spindle cells form vascular spaces, fibromas typically do not exhibit red blood cell extravasation or hyaline bodies.
Angiofibroma: While vascular spaces and red blood cell extravasation are present, hyaline bodies are not usually associated with this diagnosis.
Metastatic tumor: Cannot be completely ruled out, but requires more evidence for confirmation.
Metastatic melanoma: Cannot be completely ruled out, but requires more evidence for confirmation.
Metastatic neuroendocrine tumor: Cannot be completely ruled out, but requires more evidence for confirmation.
Metastatic lymphoma: Cannot be completely ruled out, but requires more evidence for confirmation.
Metastatic lipoma: Cannot be completely ruled out, but requires more evidence for confirmation.
Metastatic leiomyoma: Cannot be completely ruled out, but requires more evidence for confirmation.
Metastatic epithelial tumor: Cannot be completely ruled out, but requires more evidence for confirmation.
Metastatic mesenchymal tumor: Cannot be completely ruled out, but requires more evidence for confirmation.

Final Differential Diagnoses
Fibroma, Angiofibroma, Metastatic tumor, Metastatic melanoma, Metastatic neuroendocrine tumor, Metastatic lymphoma, Metastatic lipoma, Metastatic leiomyoma, Metastatic epithelial tumor, Metastatic mesenchymal tumor

Additional Examination Items
Immunohistochemistry (e.g., Vimentin, CD34, CD3, CD20, S100, CK, CD30, CD11c, CD68, fibroblastic markers, angiogenic markers, etc.)
Special histochemical stains (e.g., PAS, AB-PAS, etc.)

Molecular testing (e.g., FISH, PCR, etc.)

Summary
Differential Diagnoses: \DiffList{Fibroma, Angiofibroma, Metastatic tumor, Metastatic melanoma, Metastatic neuroendocrine tumor, Metastatic lymphoma, Metastatic lipoma, Metastatic leiomyoma, Metastatic epithelial tumor, Metastatic mesenchymal tumor}
Additional Examination Items: \ExamList{Immunohistochemistry (e.g., Vimentin, CD34, CD3, CD20, S100, CK, CD30, CD11c, CD68, fibroblastic markers, angiogenic markers, etc.), Special histochemical stains (e.g., PAS, AB-PAS, etc.), Molecular testing (e.g., FISH, PCR, etc.)}

**Follow-up Prediction:**
Re-evaluating the Initial Diagnosis
Based on the initial diagnosis, we considered the following possibilities: Fibroma, Angiofibroma, Metastatic tumor, Metastatic melanoma, Metastatic neuroendocrine tumor, Metastatic lymphoma, Metastatic lipoma, Metastatic leiomyoma, Metastatic epithelial tumor, Metastatic mesenchymal tumor.

Analyzing the Additional Test Results
Vimentin: Positive CD34: Positive S100: Negative CK: Negative CD3: Negative CD20: Negative CD30: Negative CD11c: Negative CD68: Negative PCR: Positive (for HHV-8) FISH: Positive (for HHV-8)

Further Analysis
The positivity of both Vimentin and CD34 suggests a mesenchymal origin of the tumor.
The negativity of both S100 and CK suggests it is not a neuroendocrine or epithelial tumor.
The negativity of CD3, CD20, CD30, CD11c, and CD68 suggests it is not lymphoma.
The detection of HHV-8 by both PCR and FISH suggests a possible association with Kaposi sarcoma.

Final Diagnosis
Based on the additional test results, the most likely diagnosis is Kaposi sarcoma. Kaposi sarcoma is a mesenchymal tumor caused by HHV-8, characterized by spindle cells forming vascular spaces, accompanied by red blood cell extravasation and hyaline bodies. The results of Vimentin and CD34 positivity, S100 and CK negativity, and the detection of HHV-8 by both PCR and FISH all support the diagnosis of Kaposi sarcoma. Output Final Diagnosis

\boxed{Kaposi sarcoma}

## A.12 EXTENDING TO MORE TURNS

In the main manuscript, our research centers on a two-turn diagnostic framework. However, the proposed method can naturally extend to interactions with more turns, provided that all intermediate stages remain evidence-based, i.e., they do not involve subjective outcomes such as treatment decisions.

Here, we present a representative **three-turn** diagnostic process for a case of MALT lymphoma. In this example, the additional clinical evidence is divided into two sequential stages: immunohistochemistry (IHC) results first, followed by molecular testing results. Across these three turns of interaction, the diagnostic specificity progressively increases from "lymphoma (especially bronchial lymphoma)," to "B-cell-derived lymphoma (particularly MALT lymphoma)," and ultimately to the final diagnosis of "MALT lymphoma."

| 1512 | **Case 3: MALT lymphoma (Give IHC results first, then the molecular testing)** |

**SYSTEM:**

You are Qwen, created by Alibaba Cloud. You are a helpful assistant. A conversation between User and Assistant. The user asks a question, and the Assistant solves it. The assistant first drafts the reasoning process (inner monologue) until it has derived the final answer with full confidence. It then provides a self-contained summary of the thoughts, i.e., keeping succinct but containing all the critical steps needed to reach the conclusion. It should use Markdown and Latex to format the response. Write both the thoughts and summary in the same language as the task posed by the user.

The thinking process must follow the template below (You should **include and only include one** pair of <think></think> and <answer></answer> tags in your response):

<think>The thoughts or/and draft, like working through an exercise on scratch paper. Be as casual and as long as necessary until it is confident to generate a correct answer. </think>

<answer>Here, provide a concise summary that reflects the reasoning process and presents a clear final answer to the user. </answer>

---

**USER 1:**

I need you to infer possible differential diagnoses and related additional tests based on the provided case information. Specifically, you need to analyze the given case information carefully. Then, based on the evidence points within it, gradually deduce all possible related differential diagnoses (these diagnoses must be relatively specific, avoiding simple descriptions such as "benign/malignant lesion"). Then, re-examine the case information to exclude those differential options with extremely low probability (actively reflect: does the existing information contain anything that clearly conflicts with this disease?). At the same time, you may also add new differential diagnoses (actively reflect: have I missed any possibilities?). You can engage in such reflection even during the initial inference. After repeating this process for multiple rounds, when you are sufficiently confident that the current set of differential diagnoses has a high probability of including the final diagnosis while having an extremely low likelihood of including unrelated diagnoses, you may proceed to the final summary stage. When summarizing the final differential diagnoses, rank those with a higher probability first. Note that these differential diagnoses should not exceed ten. Then, based on these differential diagnoses, provide the additional information needed to rule out some of the possible diagnoses (such as a specific immunohistochemical antibody, a specialized histochemical stain, or a particular molecular test). While thinking, you may review previous content at any time for reflection and promptly revise your conclusions as needed.

After thinking, you need to organize your response: briefly summarize your thought process, then summarize your output in the specified format.

Format requirements are as follows:

Differential Diagnoses: \DiffList{Diagnosis 1, Diagnosis 2, ...}

Further Examination Items: \ExamList{Item 1, Item 2, ...}

The following is the case information:

Male, 74 years old, presented with cough and shortness of breath. Shortness of breath occurs when climbing stairs or walking briskly. Chest CT at an outside hospital showed enlargement of the right pulmonary hilum, high-density lesions in the right main bronchus and some segmental bronchi, and patchy high-density shadows in the right lung. Pulmonary function test revealed severe mixed ventilatory dysfunction. Bronchoscopy showed irregular mucosal protrusions in the right main bronchus, upper lobe, and intermediate bronchus, causing significant luminal narrowing; a biopsy was taken from this area. Morphological features: proliferation of small, atypical monocytoid lymphocytes in the submucosa of the bronchial mucosa, with scattered immunoblasts or centroblasts present; tumor cells infiltrated the bronchial mucosal epithelium, forming lymphoepithelial lesions.

**ASSISTANT 1:**

<think> First, based on the case information, the patient is a 74-year-old male presenting primarily with cough and shortness of breath. Chest CT showed enlargement of the right pulmonary hilum, high-density lesions in the right main bronchus and some segmental bronchi, and patchy high-density shadows in the right lung. Pulmonary function tests revealed severe mixed ventilatory dysfunction. Bronchoscopy showed irregular mucosal protrusions in the right main bronchus, upper lobe, and intermediate bronchus, causing significant luminal narrowing. Biopsy revealed proliferation of small, atypical monocytoid lymphocytes in the submucosa of the bronchial mucosa, with scattered immunoblasts or centroblasts; tumor cells infiltrated the bronchial mucosal epithelium, forming lymphoepithelial lesions.

Based on this information, we can consider the following possible differential diagnoses:
1. Lymphoma (especially bronchial lymphoma)
2. Adenocarcinoma
3. Sarcoma
4. Fibrous tumor
5. Lymphocytosis
6. Mucinous tumor
7. Metastatic carcinoma
8. Inflammatory diseases (e.g., interstitial lung disease)
9. Benign hyperplasia (e.g., adenoma)
10. Lymphoepithelial lesions

Next, we re-evaluate these diagnoses and exclude those with very low probability. By analyzing the case information, we find that the submucosal proliferation of small, atypical monocytoid lymphocytes with scattered immunoblasts or centroblasts, along with tumor cell infiltration of the bronchial mucosal epithelium forming lymphoepithelial lesions, suggests lymphoma or an inflammatory disease. Adenocarcinoma, sarcoma, fibrous tumor, and mucinous tumor are less likely because the case does not show obvious features of these conditions. Benign hyperplasia is also unlikely due to the presence of clearly abnormal cell proliferation. The possibility of metastatic carcinoma cannot be completely excluded, as a tumor from another site could have metastasized to the lung. Therefore, we can narrow down the differential diagnosis to the following more likely possibilities:
1. Lymphoma (especially bronchial lymphoma)
2. Inflammatory diseases (e.g., interstitial lung disease)
3. Lymphocytosis
4. Metastatic carcinoma
5. Lymphoepithelial lesions

Upon reflection, we note that lymphoepithelial lesions represent a specific diagnosis that may result from abnormal proliferation of lymphocytes and epithelial cells. Therefore, we can further narrow the differential diagnosis to the following likely possibilities:
1. Lymphoma (especially bronchial lymphoma)
2. Inflammatory diseases (e.g., interstitial lung disease)
3. Lymphocytosis
4. Metastatic carcinoma
5. Lymphoepithelial lesions
</think>
<answer>
**Differential diagnosis:** \DiffList{Lymphoma (especially bronchial lymphoma), Inflammatory diseases (e.g., interstitial lung disease), Lymphocytosis, Metastatic carcinoma, Lymphoepithelial lesions}
**Further Examination Items:** \ExamList{Immunohistochemistry (e.g., CD3, CD20, CK, S100, Vimentin, MUC1, MUC5AC, MUC6, CD11c, CD68, CD23, CD15, CD11b, CD1a, CD30, CD138, etc.), special stains (e.g., PAS, AB-PAS, etc.), molecular testing (e.g., gene expression analysis of BCL2, BCL6, MALT1, TP53, KRAS, NRAS, BRAF, etc.)} </answer>

1620
1621
1622
1623
1624
1625
1626
1627
1628
1629
1630
1631
1632
1633
1634
1635
1636
1637
1638
1639
1640
1641
1642
1643
1644
1645
1646
1647
1648
1649
1650
1651
1652
1653
1654
1655
1656
1657
1658
1659
1660
1661
1662
1663
1664
1665
1666
1667
1668
1669
1670
1671
1672
1673

**USER 2:**

Now some of the additional test results are available. I need you to:
1. First review the "case information" and the "first-round diagnosis" to clarify the previous diagnostic reasoning and conclusions.
2. Then review the "additional test results." The additional results may not fully correspond to the extra tests requested during the first-round diagnosis. Based on the available results, you should further refine the differential diagnosis on top of the first-round assessment, and provide both the diagnosis and any additional tests that should be performed next. Note: after receiving more information, you may completely overturn the first-round reasoning and provide a diagnosis based solely on the current evidence.
3. The final output format must follow:
**Differential Diagnosis:** \DiffList{Diagnosis 1, Diagnosis 2, ... }
**Further Examination Items:** \ExamList{Item 1, Item 2, ... }

**Additional Test Results**
CD3: Negative
CD20: Positive
CK: Negative
S100: Negative
Vimentin: Positive
MUC1: Negative
MUC5AC: Negative
MUC6: Negative
CD11c: Positive
CD68: Positive
CD23: Negative
CD15: Negative
CD11b: Negative
CD1a: Negative
CD30: Negative
CD138: Positive

---

**ASSISTANT 2:**

<think>First, we review the diagnostic reasoning and conclusions from the first round. The initial diagnostic considerations included the following relatively likely possibilities: lymphoma (especially bronchial lymphoma), inflammatory diseases (such as interstitial lung disease), lymphocytosis, metastatic carcinoma, and lymphoepithelial lesions. According to the morphological description, there is proliferation of mononuclear, atypical, clear small lymphocytes beneath the bronchial mucosa, with scattered immunoblasts or centroblasts among them; tumor cells infiltrate the bronchial mucosal epithelium, forming lymphoepithelial lesions. These findings suggest the possibility of lymphoma or inflammatory disease. Adenocarcinoma, sarcoma, fibrous tumors, mucinous tumors, and other such entities are relatively less likely because the case information does not strongly support these features. Benign hyperplasia is also unlikely due to the presence of clearly abnormal cellular proliferation. Metastatic carcinoma cannot be entirely ruled out, as tumors from other sites may metastasize to the lung. Therefore, we can narrow the diagnostic range to the following relatively likely diagnoses: lymphoma (especially bronchial lymphoma), inflammatory diseases (such as interstitial lung disease), lymphocytosis, metastatic carcinoma, and lymphoepithelial lesions.
Next, we examine the additional test results. These include immunohistochemical staining and special staining results. The specific results are as follows: - CD3: negative - CD20: positive - CK: negative - S100: negative - Vimentin: positive - MUC1: negative - MUC5AC: negative - MUC6: negative - CD11c: positive - CD68: positive - CD23: negative - CD15: negative - CD11b: negative - CD1a: negative - CD30: negative - CD138: positive
Based on these results, we can analyze further. CD20 and CD138 positivity suggests a B-cell-derived neoplasm, such as diffuse large B-cell lymphoma or MALT lymphoma. CD3 nega-

tivity, CD20 positivity, and CD138 positivity point toward B-cell lymphoma. CD68 positivity suggests a histiocytic component. CD11c and CD68 positivity indicates possible histiocytic or macrophage lineage. CD23 negativity, CD15 negativity, CD11b negativity, CD1a negativity, and CD30 negativity suggest a non-Hodgkin lymphoma rather than Hodgkin lymphoma. The negativity of MUC1, MUC5AC, and MUC6 suggests a non-mucinous tumor. Taken together, the most likely diagnosis is a B-cell-derived lymphoma, particularly MALT lymphoma.

Based on this new information, we can further narrow the differential diagnosis to the following relatively likely possibilities: B-cell-derived lymphoma (particularly MALT lymphoma), inflammatory diseases (such as interstitial lung disease), and metastatic carcinoma. Other less likely diagnoses need not be considered further, as the new test results provide more information. </think>

<answer>

**Differential diagnosis:** \DiffList{B-cell-derived lymphoma (particularly MALT lymphoma), inflammatory diseases (such as interstitial lung disease), metastatic carcinoma}

**Further Examination Items:** \ExamList{Molecular testing (such as BCL2, BCL6, MALT1, TP53, KRAS, NRAS, BRAF and other gene expression analyses)}

</answer>

## USER 3:

Now the results of the molecular testing are available. I need you to: 1. First review the "case information" and the previous rounds of diagnoses to clarify the earlier diagnostic reasoning and related conclusions. 2. Then review this round of "additional test results." The additional test results may not fully match the extra tests that were previously requested. Based on the available results, you should further analyze the differential diagnosis on top of the previous conclusions and provide the final diagnosis. Note: after receiving new information, you may completely overturn the prior diagnostic reasoning and provide the diagnosis supported by the current data. 3. The final diagnosis must be output in the specified format, e.g.: \boxed{Diagnosis}

**Additional Test Results:**
BCL2: no rearrangement
BCL6: no rearrangement
MALT1: rearrangement positive
TP53: no mutation
KRAS: no mutation
NRAS: no mutation
BRAF: no mutation

## ASSISTANT 3:

<think>First, we review the diagnostic reasoning and conclusions from the previous rounds. The earlier differential diagnosis focused on the following most likely possibilities: B-cell lymphoma (particularly MALT lymphoma), inflammatory diseases (such as interstitial lung disease), and metastatic carcinoma. Based on the morphological description and immunohistochemical findings-namely, the proliferation of monocytoid, atypical clear small lymphocytes in the bronchial submucosa, scattered immunoblasts or centroblasts, and tumor cell infiltration of the bronchial mucosal epithelium forming lymphoepithelial lesions—these features suggested lymphoma or inflammatory disease. Among them, B-cell lymphoma, especially MALT lymphoma, was the most likely diagnosis. CD20 and CD138 positivity indicated a B-cell neoplasm such as MALT lymphoma. CD68 positivity suggested a possible histiocytic component. CD11c and CD68 positivity indicated histiocytes or macrophages. CD23 negativity, CD15 negativity, CD11b negativity, CD1a negativity, and CD30 negativity supported non–Hodgkin lymphoma rather than Hodgkin lymphoma. The negativity for MUC1, MUC5AC, and MUC6 indicated a non-mucinous tumor.

Next, we review the additional test results from this round. These included molecular findings. The specific results are as follows: - BCL2: no rearrangement - BCL6: no rearrangement

- MALT1: rearrangement positive - TP53: no mutation - KRAS: no mutation - NRAS: no mutation - BRAF: no mutation

Based on these results, we can further analyze the diagnosis. The absence of BCL2 and BCL6 rearrangements suggests a lymphoma without BCL2 or BCL6 rearrangements. A positive MALT1 rearrangement strongly supports MALT lymphoma. The absence of TP53, KRAS, NRAS, and BRAF mutations suggests a lymphoma without these genetic mutations. In summary, the most likely diagnosis is MALT lymphoma.

Based on this new information, we can determine that the final diagnosis is MALT lymphoma.

</think>

<answer> \boxed{MALT lymphoma} </answer>

