# OpenReview forum: "Reinforcement Learning for Evidence-Seeking Diagnostic Reasoning with Large Language Models"
_ICLR.cc/2026/Conference — ICLR 2026 Conference Withdrawn Submission_

### Official Review · Reviewer_hXAC · 2025-10-15

**Soundness:** 3
**Presentation:** 3
**Contribution:** 3
**Rating:** 6
**Confidence:** 3

**Summary:**

This paper formalizes medical diagnosis as a two-turn, evidence-seeking process to better mimic real-world clinical reasoning. A reinforcement learning framework guided by three novel rewards is proposed, including format, rank-sensitive diagnosis, and examination consistency, to train LLMs to request and utilize evidence effectively. To facilitate this process, the authors introduce RAGES, a retrieval-augmented system for simulating realistic follow-up examination results. Experiments on multilingual datasets demonstrate that LLMs significantly improve diagnostic accuracy with additional evidence and that the proposed model is competitive with larger, reasoning-enhanced baselines.

**Strengths:**

1. The paper formulates diagnosis as an interactive, two-turn evidence-seeking task, moving beyond the prevalent single-turn evaluation paradigm, which is clinically relevant.
2. The rank-sensitive diagnosis reward is designed to encourage broader differentials initially and precise diagnoses later, with a dynamic hyperparameter strategy that adapts to the amount of available evidence.
3. The proposed RAGES can generate plausible evidence by combining real data, a structured knowledge base, and LLM generation, which is shown to be more effective than pure generation in the ablation experiments.

**Weaknesses:**

1. While a two-round diagnosis is iterative and can gather more information, there are also cases where two rounds are not enough for diagnosis or one round is enough. The proposed framework and its design are only applicable to the two-round setting, which limits the broader applicability.
2. Only three datasets are evaluated, and one of them is in-house. Moreover, the dataset sizes are quite small, so it’s unclear whether the models can generalize well and retain their original medical capabilities e.g., in standard zero-shot medical QA tasks.
3. There is room for improvement in the presentation. It would be clearer to first provide an overview of the overall framework: explaining how the model learns to perform Turn 1, how evidence is generated between turns (RAGES), and how it learns to perform Turn 2, before diving into the specific steps.

**Questions:**

Since there are only 959 instances in the training set, the fact that training took 40 hours on 8 H100 GPUs seems quite long. Is the bottleneck mainly due to the LLM-as-a-judge during training?

---

> ### Author Response · Authors · 2025-11-20
>
> We sincerely thank you for your valuable feedback and encouraging score. We have updated our manuscript accordingly. Modified text is highlighted in blue, and for any newly added sections, only the section titles are colored for clarity and visual distinction. Below, we provide detailed responses to each of your comments.
>
> ---
>
> **W1. Two-turn paradigm and broader applicability**
>
> Thank you for pointing this out. We acknowledge that some real-world cases may require more than two turns, while others may be resolved in one. We would like to clarify that our framework and training methodology are **not inherently limited to two turns**.
>
> The two-turn design was chosen to model a common clinical pattern: an initial differential diagnosis followed by evidence-seeking to refine the conclusion. In practice, our framework can be extended to multiple turns by repeatedly invoking the differential diagnosis and evidence-request prompts, allowing the model to continue gathering and integrating information. We added a **three-turn example** in **Appendix A.12**, illustrating how successive evidence (IHC results first, then molecular testing results) leads to progressively more specific diagnoses (from “lymphoma” → “B-cell-derived lymphoma” → “MALT lymphoma”).
>
> ---
>
> **W2. Dataset scope and generalization**
>
> Thank you for this suggestion. To assess generalization and whether the models retain original medical capabilities, we conducted experiments on the **MedQA dataset**, including both English and Simplified Chinese test sets. Results are summarized below (**Appendix A.4** provides full details):
>
> | Model        | English (1273) | Simplified Chinese (3426) |
> | ------------ | -------------- | ------------------------- |
> | Qwen2.5-7B   | 53.5           | 82.8                      |
> | Qwen2.5-32B  | 68.5           | 89.7                      |
> | Ours-RL-7B   | 52.5           | 82.9                      |
> | Ours-SFT-32B | 74.0           | 87.1                      |
>
> The results show little degradation in zero-shot medical QA performance, indicating that task-specific training does not compromise original medical capabilities. The lack of clear improvement may be due to the limited data size and focus on a single diagnosis task.
>
> ---
>
> **W3. Presentation and framework overview**
>
> Thank you for the suggestion. We revised **Section 3.1** to provide a clearer high-level overview of our framework:
>
> *... This two-turn paradigm is established with three key components. First, we utilize real-world diagnostic cases collected from multiple sources to provide plentiful information for learning (Section 3.4). Second, our proposed RL framework trains the model to produce reasonable differential diagnoses and to actively request further evidence (Section 3.2). Third, the RAGES method connects different turns by generating simulated test results to guide subsequent reasoning (Section 3.3)...*
>
> ---
>
> **Q1. Training time with only 959 instances**
>
> Thank you for noticing this. We discuss this in **Appendix A.6**, and the main factors are:
>
> 1. **Inefficient GPU usage with LLM judges:** OpenRLHF efficiently schedules multiple models, but integrating a judge model that directly returns text was not feasible at that time. We dedicated 4 GPUs to the judge and transferred information via HTML, which reduces efficiency.
> 2. **Computational cost of LLM judges:** Each generation requires the judge twice (once for diagnostic evaluation and once for examination plausibility), effectively doubling cost. Skipping the examination judge reduces training to ~28 hours. Using a smaller judge (Qwen2.5-7B) lowers costs but introduces counting and judging errors, so we used Qwen2.5-32B on 4 GPUs despite inefficiency.
>
> ---
>
> In the end, we want to thank you again for you valuable suggestions. Your suggestions help us present our idea in a more reader-friendly way, validate generalization and retention of medical capabilities via MedQA experiments and notice the long training time problem.

---

> > ### Comment · Reviewer_hXAC · 2025-11-25
> >
> > Thank you for your clarification and additional experiments. I'd like to maintain my borderline score that there're some strengths in the paper such as the formulation of two-turn interactive while also some limitations like inefficiency and somewhat limited evaluation.

---

> > > ### Author Response · Authors · 2025-12-03
> > >
> > > Thanks for your reply.

---

### Official Review · Reviewer_rvSr · 2025-10-27

**Soundness:** 3
**Presentation:** 2
**Contribution:** 3
**Rating:** 4
**Confidence:** 3

**Summary:**

This paper introduces a reinforcement learning (RL) framework designed to enhance the evidence-seeking diagnostic reasoning of large language models (LLMs). Recognizing that real-world medical diagnosis often requires iterative evidence collection, the authors propose a two-turn diagnostic paradigm where an LLM first generates differential diagnoses and suggests additional tests, then refines its reasoning after receiving simulated results.
The study introduces:

Diagnostic evidence-seeking rewards—comprising format, rank-sensitive diagnosis, and examination consistency rewards—to guide structured and clinically valid reasoning.

RAGES (Retrieval-Augmented Generation-based Examination Simulation)—a method that synthesizes realistic follow-up evidence by reusing, retrieving, and generating examination results.
Empirical results show improved diagnostic accuracy and plausibility compared to strong baselines, highlighting the potential of RL-based reasoning for interactive medical diagnosis.

**Strengths:**

**Novel framing of multi-turn diagnosis**:
The paper explicitly formalizes diagnostic reasoning as a two-turn, evidence-seeking process. This mirrors real clinical workflows, bridging a critical gap between static, single-turn evaluations and dynamic reasoning processes.

**Well-structured reward design**:
The combination of rank-sensitive, format, and consistency rewards is both theoretically grounded and practically effective. The adaptive τ mechanism for balancing exploration (broad differential lists) and exploitation (precise diagnosis) demonstrates thoughtful design.

**Comprehensive evaluation**:
The experiments span multilingual datasets (Chinese and English), use multiple strong LLM judges (DeepSeek-R1, Qwen2.5-Max, GPT-5), and provide clear metrics for differential and final diagnostic accuracy. The ablation studies further validate design choices.

**Weaknesses:**

**Limited model scale and comparison scope**:
The proposed RL model (7B parameters) performs well in early reasoning but lags behind larger reasoning-tuned models (e.g., QwQ-32B, Qwen3-32B) in final diagnosis. This may limit its practical competitiveness, especially for high-stakes applications.

**Offline RAGES simulation**:
Since RAGES operates offline rather than interactively during training, the framework cannot fully exploit real-time feedback or reinforcement from generated evidence. This reduces the fidelity of multi-turn reasoning in training loops.

**Reward interpretation complexity**:
Although theoretical analysis of τ and reward functions is provided, the practical intuition behind how reward shaping influences diagnostic reasoning behavior could be better illustrated with concrete examples or case visualizations.

**Data generalizability and reproducibility**:
The dataset includes in-house Chinese pathology cases not publicly released, limiting replication. Moreover, while ethical considerations are discussed, it’s unclear whether the model generalizes beyond pathology to broader medical domains.

**Questions:**

**Dynamic integration of RAGES**:
Could RAGES be integrated within the RL training process (rather than offline) to enable genuine multi-turn reinforcement? How would this impact model stability and computation?

**Human-in-the-loop evaluation**:
While the study uses LLM judges, have the authors considered validation by medical experts to ensure alignment with real clinical reasoning standards?

**Scalability of the reward framework**:
The proposed rank-sensitive and consistency rewards are domain-specific. How adaptable is this framework to other domains (e.g., radiology or symptom-based triage) where evidence is less structured?

**Safety and interpretability considerations**:
Given the model’s diagnostic nature, how does the RL framework ensure safe, interpretable outputs—especially when RL-driven optimization might amplify confident but incorrect reasoning paths?

---

> ### Author Response · Authors · 2025-11-20
>
> We sincerely thank you for your detailed suggestions. We have updated our manuscript accordingly. Modified text is highlighted in blue, and for any newly added sections, only the section titles are colored for clarity and visual distinction. Below, we provide detailed responses to each of your comments.
>
> ---
>
> **W1. Limited model scale and comparison scope**
>
> Thank you for your comment. We acknowledge that our 7B RL model lags behind larger reasoning-tuned models (e.g., QwQ-32B, Qwen3-32B) in final diagnosis accuracy. Inspired by Reviewer RqfJ, we conducted additional experiments with similar-sized models (**HuatuoGPT-o1-7B** and **Ours-SFT-7B**) in **Appendix A.5**. The results are summarized below:
>
> | Model       | DiffAcc (EN) | DiffAcc (CN) | DiffAcc (Mean) | DxAcc (EN) | DxAcc (CN) | DxAcc (Mean) |
> | ----------- | ------------ | ------------ | -------------- | ---------- | ---------- | ------------ |
> | Huatuo-7B   | 31.5         | 40.1         | 37.7           | 19.7       | 22.5       | 21.7         |
> | Ours-SFT-7B | 45.5         | 51.2         | 49.6           | 29.4       | 30.4       | 30.1         |
> | Ours-RL-7B  | 62.0         | 65.8         | 64.8           | 36.4       | 39.0       | 38.2         |
>
> These results demonstrate that the proposed RL method achieves the best performance among models of similar size, validating its potential.
>
> ---
>
> **W2. Offline RAGES simulation & Q1. Dynamic integration of RAGES**
>
> Thank you for this suggestion. We acknowledge the limitation of offline simulation and we did consider integrating RAGES into the RL loop, but simulating examination results online is computationally heavier than evaluating diagnoses, as it requires a stronger model. Our current setup already dedicates **Qwen2.5-32B** for both diagnosis and examination plausibility evaluation, which heavily impacts training efficiency (see **Appendix A.6**). Using a stronger model for real-time simulation would require substantially more resources. Meanwhile, simulating multi-turn rollouts is also important with the RL framework. We are now approaching VeRL, but only at the very beginning stage.
>
> ---
>
> **W3. Reward interpretation complexity**
>
> We are thankful for this suggestion. We now include a new figure (current **Fig. 3**) visualizing the rank-sensitive diagnostic reward and updated the discussion in **Section 5.4** to more clearly illustrate how reward shaping guides diagnostic reasoning.
>
> ---
>
> **W4. Data generalizability and reproducibility**
>
> Thanks for pointing this out. We will release URLs for all public cases to promote reproducibility. For generalizability, inspired by Reviewer hXAC, we present **MedQA** results in **Appendix A.4**. A “TL;DR” summary shows no general improvement but also no degradation, likely due to the single diagnosis task and limited data.
>
> | Model        | English (1273) | Simplified Chinese (3426) |
> | ------------ | -------------- | ------------------------- |
> | Qwen2.5-7B   | 53.5           | 82.8                      |
> | Qwen2.5-32B  | 68.5           | 89.7                      |
> | Ours-RL-7B   | 52.5           | 82.9                      |
> | Ours-SFT-32B | 74.0           | 87.1                      |
>
> ---
>
> **Q2. Human-in-the-loop evaluation**
>
> We agree that involving medical experts is important. However, given practical constraints on time and cost, we currently rely on multi-model evaluation as a **preliminary proxy**. We note this as a limitation and plan to include human evaluation in future work (see discussion **lines 530–532**).
>
> ---
>
> **Q3. Scalability of the reward framework**
>
> We appreciate this insightful question. The rank-sensitive and consistency rewards are generalizable to other domains as long as evidence stages and desired outcomes are defined. Additionally, the concept of a ranked differential diagnosis is broadly applicable across medical domains, making the reward framework potentially transferable. We discuss this as a future direction (**lines 526–529**). When focusing on the situations given, we think they represents two interesting cases:
>
> - **Radiology**: Often resembles a single **evidence collection** stage; other models can be applied to interpret imaging results and provide the final observation.
> - **Symptom-based triage**: Represents a preliminary stage before diagnosis, with less structured evidence. Our framework could accommodate such cases by defining intermediate outputs and potential candidate outcomes.

---

> > ### Author Response · Authors · 2025-11-20
> >
> > **Q4. Safety and interpretability considerations**
> >
> > We agree that safe and interpretable outputs are important. Our current work focuses on enabling models to actively request information and revise diagnoses, not yet on safety mechanisms. We discuss this explicitly as future work (**lines 530–532**). Two complementary paths can be pursued: (1) existing methods like PRM and (2) human-in-the-loop strategies (e.g., DPO) to mitigate incorrect high-confidence outputs.
> >
> > ---
> >
> > We are sincerely grateful for your insightful comments. They help us (1) visualize and interpret the reward function more clearly; (2) conduct more comprehensive experiments across models of similar size; (3) clarify future work on broader domain applicability, and safety-oriented framework design.

---

> > ### Comment · Reviewer_rvSr · 2025-11-20
> >
> > Thank you for your detailed response for my interests.
> >
> > > W1: We acknowledge that our 7B RL model lags behind larger reasoning-tuned models (e.g., QwQ-32B, Qwen3-32B) in final diagnosis accuracy.
> >
> > The reviewer understands the difficulty of conducting extensive RL experiments within a tight revision cycle. However, the reviewer still recommends extending the RL experiments in the final revision, if possible, to further demonstrate the effectiveness of your RL approach. Even limited-scale additional experiments—such as longer training horizons or expanded trajectory sets—would help strengthen the empirical claims.
> >
> > > W2. Offline RAGES simulation & Q1. Dynamic integration of RAGES
> >
> > Most current RL frameworks for LLMs are actively moving toward multi-turn interactions and online environment feedback. In this regard, your direction toward verl and multi-turn rollouts is aligned with the field’s trajectory. While the reviewer understands the computational constraints that prevent full integration of real-time RAGES simulation at this stage, the reviewer encourages the authors to elaborate more clearly in the revision on how the proposed method can be extended in the future toward online or partially online simulation. Clarifying this roadmap would help readers better appreciate the long-term potential of your approach.
> >
> > > W3. Reward interpretation complexity
> >
> > The reviewer finds that the results presented in Figure 3 effectively illustrate the key theoretical insights and support the main findings. The visual summary captures the reward structure well, and I believe it will help readers grasp the core ideas more intuitively.
> >
> > ---
> > Many of the remaining issues raised in the author's response understandably cannot be fully resolved within the current revision stage due to computational and resource limitations. Given the current scope and constraints, I believe the authors have responded reasonably. Therefore, I would like to maintain my original score.

---

> > > ### Author Response · Authors · 2025-12-03
> > >
> > > Sorry for the late reply. We are working on providing more empirical findings. After our initial responses, we urgently sought computing resources and attempted to train Qwen2.5-32B-Instruct. After multiple trials, we managed to begin training with 16x H200 GPUs (RL) and 8x H100 GPUs (LLM judge). The training was expected to take over 120 hours due to slower rollout generation and inter-machine communication. Unfortunately, training only reached about 30%, so we cannot provide final evaluation results. However, we observed that early checkpoints already outperform the base model, suggesting strong potential. We paste the partial results below.
> > >
> > > | Model       | DiffAcc (EN) | DiffAcc (CN) | DiffAcc (Mean) | DxAcc (EN) | DxAcc (CN) | DxAcc (Mean) |
> > > | ----------- | ------------ | ------------ | -------------- | ---------- | ---------- | ------------ |
> > > | Qwen-32B    | 42.7         | 51.9         | 49.3           | 24.5       | 34.2       | 31.4         |
> > > | Ours-RL-32B | 45.8         | 60.1         | 56.0           | 32.1       | 43.9       | 40.5         |
> > >
> > > We don't know whether you can see this late reply, but we sincerely thank you for your valuable feedback and try our best to present what we see to you.

---

### Official Review · Reviewer_RqfJ · 2025-11-01

**Soundness:** 2
**Presentation:** 3
**Contribution:** 2
**Rating:** 4
**Confidence:** 4

**Summary:**

This work designs a two-turn diagnostic paradigm for medical diagnosis and proposes a new reinforcement learning reward to train a diagnosing reasoning model.

**Strengths:**

1. The paper is well written, and the formulas and figures help with understanding.

2. The paper has a careful design of the two-turn diagnostic paradigm, the reinforcement learning reward, and RAGES. It proposes a novel architecture for medical diagnosing.

3. The proposed method improves the model over both reasoning and non-reasoning baselines.

4. The paper designs ablation studies to verify the effectiveness of the reward design and RAGES.

**Weaknesses:**

1. All evaluations are performed in a two-turn setting. There is no strong evidence that the proposed two-turn diagnosis has advantages over the traditional one-turn diagnosis.

2. The proposed two-turn diagnosis is a new paradigm, but it seems to consume more compute. It requires 40 hours of training on 8xH100 with only ~1k data. And there is no analysis on inference efficiency.

3. The DiffAcc of Ours-RL-7B in Table 1 seems good, while the DxAcc of Ours-RL-7B in Table 2 shows a relatively small advantage. Does this mean that the second turn in your designed dialogue fails to find out the correct disease?

**Questions:**

1. You could provide some evaluation results of traditional one-turn diagnosis to further prove the advantage of the proposed two-turn diagnosis method.

2. The DxAcc of Ours-RL-7B only surpasses the non-reasoning Qwen2.5-32B baseline in Table 2. You could provide more comparisons with other diagnostic models (e.g., HuatuoGPT-o1) to prove that your model is stronger.

3. You could provide more analysis on the inference efficiency and training efficiency.

4. You could give a more in-depth analysis on the DxAcc of Ours-RL-7B. It would be better if you provide a fair comparison between Ours-RL-7B and `Ours-SFT-7B'.

---

> ### Author Response · Authors · 2025-11-20
>
> We sincerely thank you for your constructive guidance. We have updated our manuscript accordingly. Modified text is highlighted in blue, and for any newly added sections, only the section titles are colored for clarity and visual distinction. Below, we provide detailed responses to each of your comments.
>
> ---
>
> **W1 / Q1. Evidence for advantage of two-turn diagnosis over one-turn**
>
> Thank you for this valuable suggestion. We agree that comparing with a traditional one-turn diagnosis is important to demonstrate the benefit of our two-turn design.
>
> In **Section 5.1 (“More Evidence, More Accurate Diagnosis”)**, we already show that the main diagnosis consistently improves after acquiring additional evidence. To further illustrate, we conducted an **idealized one-turn experiment**: models are provided with all available information upfront and asked for a direct diagnosis, simulating a traditional one-turn approach with oracles. Results are summarized below (see **Appendix A.3** for a visual version):
>
> | Model        | Direct DxAcc (EN) | Direct DxAcc (CN) | Evidence-seeking DxAcc (EN) | Evidence-seeking DxAcc (CN) | Evidence-seeking DiffAcc (EN) | Evidence-seeking DiffAcc (CN) |
> | ------------ | ----------------- | ----------------- | --------------------------- | --------------------------- | ----------------------------- | ----------------------------- |
> | Qwen2.5-32B  | 33.9              | 31.7              | 24.5 ↓9.4                   | 34.2 ↑2.5                   | 42.7 ↑8.8                     | 51.9 ↑20.0                    |
> | Qwen2.5-72B  | 43.3              | 37.3              | 43.9 ↑0.6                   | 45.4 ↑8.0                   | 54.2 ↑11.0                    | 65.6 ↑28.0                    |
> | QwQ-32B      | 37.3              | 43.7              | 42.7 ↑5.4                   | 50.4 ↑6.7                   | 57.6 ↑20.0                    | 66.1 ↑22.0                    |
> | Qwen3-32B    | 36.0              | 42.7              | 29.7 ↓6.3                   | 43.1 ↑0.4                   | 44.3 ↑8.3                     | 56.3 ↑14.0                    |
> | Huatuo-7B    | 22.4              | 24.0              | 19.7 ↓2.7                   | 21.1 ↓2.9                   | 31.5 ↑9.1                     | 40.1 ↑16.0                    |
> | M2-32B       | 37.6              | 44.3              | 37.9 ↑0.3                   | 49.2 ↑4.9                   | 52.7 ↑15.0                    | 66.0 ↑22.0                    |
> | Ours-SFT-32B | 40.9              | 43.6              | 42.7 ↑1.8                   | 51.3 ↑7.7                   | 54.9 ↑14.0                    | 67.0 ↑23.0                    |
> | Ours-RL-7B   | 29.1              | 34.4              | 36.4 ↑7.3                   | 39.0 ↑4.6                   | 62.0 ↑33.0                    | 65.8 ↑31.0                    |
>
> Here are several short analysis:
>
> - Evidence-seeking (two-turn) improves diagnostic accuracy for most models.
> - Some models (e.g., Qwen2.5-32B, Qwen3-32B) perform slightly better in one-turn DxAcc for English cases, likely because these cases contain sufficient and guiding information upfront.
> - Huatuo-7B shows degraded performance, reflecting strong adherence to its one-turn training patterns.
> - The differential accuracy signals substantial improvement potential, emphasizing the value of sequential evidence acquisition.
>
> ---
>
> **W2 / Q3. Compute cost and inference efficiency**
>
> We appreciate the comment regarding training and inference efficiency. We have added a detailed discussion in **Appendix A.6**. Here is a brief discussion:
>
> 1. **Inefficient GPU usage:** OpenRLHF efficiently allocates GPUs, but integrating a judge model that returns text directly was not feasible. We dedicated 4 GPUs to the judge and used HTML-based communication, which reduces efficiency.
> 2. **High cost of LLM judges:** Each generation requires the judge twice: for diagnosis and examination plausibility. Skipping the examination judge reduces training from ~40h to ~28h. Using a smaller judge (Qwen2.5-7B) cuts costs but introduces errors, so we used Qwen2.5-32B on 4 GPUs despite inefficiency.
>
> **Inference efficiency:** See Appendix A.6 for a quantitative evaluation. In brief, additional reasoning (second turn) increases token generation and slows inference, but the improvement in accuracy justifies the extra cost.

---

> > ### Author Response · Authors · 2025-11-20
> >
> > **W3 / Q2 / Q4. DxAcc and comparison with other models**
> >
> > Thank you for highlighting this point. We further compare **Ours-RL-7B** with **HuatuoGPT-o1-7B** and **Ours-SFT-7B** in **Appendix A.5**.  Here is a copy of the main results.
> >
> > | Model       | DiffAcc (EN) | DiffAcc (CN) | DiffAcc (Mean) | DxAcc (EN) | DxAcc (CN) | DxAcc (Mean) |
> > | ----------- | ------------ | ------------ | -------------- | ---------- | ---------- | ------------ |
> > | Huatuo-7B   | 31.5         | 40.1         | 37.7           | 19.7       | 22.5       | 21.7         |
> > | Ours-SFT-7B | 45.5         | 51.2         | 49.6           | 29.4       | 30.4       | 30.1         |
> > | Ours-RL-7B  | 62.0         | 65.8         | 64.8           | 36.4       | 39.0       | 38.2         |
> >
> > From the results, we can see that Ours-RL-7B consistently outperforms models of similar size in both differential and final diagnosis accuracy. While developing a larger-scale “Ours-RL-32B” is challenging, the trend suggests strong potential for the RL approach.
> >
> > ---
> >
> > In the end, we would like to show our  sincere gratitude again for your constructive suggestions. They help us better justify the evidence-seeking paradigm with both idealized and differential analysis, clarify sources of high training cost and report inference efficiency, and conduct more comprehensive evaluations with additional baseline and SFT models.

---

> > > ### Comment · Reviewer_RqfJ · 2025-11-25
> > > **Thanks for the Rebuttal**
> > >
> > > Thanks the authors for providing the rebuttal. Most of my concerns have been addressed. I will raise my score to 6 accordingly.

---

> > > > ### Author Response · Authors · 2025-12-03
> > > >
> > > > Thanks for your reply and score change!

---

### Official Review · Reviewer_Wi1g · 2025-11-01

**Soundness:** 3
**Presentation:** 3
**Contribution:** 3
**Rating:** 4
**Confidence:** 4

**Summary:**

The paper studies evidence-seeking diagnostic reasoning with LLMs as a two-turn process trained with reinforcement learning. The method uses three reward signals that can be checked automatically. The first enforces a structured output. The second is a rank-sensitive diagnosis reward that scores the position of the correct disease in the differential list and adapts with a temperature-like parameter tau. The third checks whether requested tests are consistent with the candidate diseases. The paper also introduces RAG-based Examination Simulation, called RAGES, which blends reuse of real case results, retrieval from a disease–exam knowledge base, and generation to produce follow-up evidence. Experiments on English and Chinese pathology cases report higher accuracy after the second turn and competitive results against stronger or larger baselines. The ablations show how tau affects list length and accuracy and how each RAGES component contributes to output quality.

**Strengths:**

1. The work gives a clear and formal treatment of evidence-seeking diagnosis that goes beyond the common single-turn setup in medical LLM research. The framework is two-turn by design and ties learning targets to verifiable signals, which helps analysis and training.

2. The definition of a rank-sensitive reward is careful and well argued. The paper proves two theorems that show how earlier hits and shorter lists receive higher scores and provides findings that explain how tau changes behavior across stages. The ablation with different tau settings makes these points concrete.

**Weaknesses:**

1. The method fixes diagnosis into a two-turn script, which narrows external validity for clinical work that often needs many turns. With only one round of evidence requests and one round of updates, the study cannot test longer-horizon skills such as sequential test planning with cost and risk, recovery from misleading results, or non-myopic revision over several cycles. This choice flows into dataset design and reward choice, which may tune the policy to a staged pattern and leaves open whether behavior would hold when three, five, or ten turns of new evidence arrive.


2. This paper is an end-to-end reliance on LLM judges for both optimization and reporting. During reinforcement learning, an LLM judge decides hits, ranks the ground-truth position for the rank-sensitive reward, and assigns the examination-plausibility bonus. The same family of models is also used at evaluation time to score differential accuracy and final diagnosis accuracy, and to assess the plausibility of examinations. Without human adjudication or agreement checks, it is hard to separate true gains in diagnostic skill from alignment with judge preferences or reward hacking.


3. The examination simulation lacks clinical validation. RAGES composes follow-up results by reusing overlapping findings, retrieving disease–exam mappings from a structured knowledge base, and prompting a strong model to fill in the rest. The quality of these simulated results and the plausibility of requested tests are judged by an LLM rather than by clinicians or by comparison with real test outcomes, which makes it unclear whether the gains reflect clinically sound choices or model-to-model agreement.

**Questions:**

Please see the weakness above.

---

> ### Author Response · Authors · 2025-11-20
>
> We sincerely thank you for your insightful reviews. We have updated our manuscript accordingly. Modified text is highlighted in blue, and for any newly added sections, only the section titles are colored for clarity and visual distinction. Below, we provide detailed responses to each of your comments.
>
> ---
>
> **W1. Whether the proposed two-turn fashion can accommodate more complex, multi-turn medical consultations?**
>
> Thank you for your valuable feedback. We fully agree that longer-horizon planning and multi-turn interactions are crucial in real-world clinical practice. We would like to clarify that our overall training strategy and framework are **not inherently limited to two turns**.
>
> Our RL policy is based on a general principle: when information is insufficient, clinicians often propose a ranked differential diagnosis. To model this, we introduced a **rank-sensitive diagnostic reward**. In the current experiments, we use two prompts: the first for a ranked differential diagnosis with recommended tests, and the second for a final diagnosis. Importantly, this design does **not preclude multi-turn reasoning**. By repeatedly invoking the differential diagnosis prompt, our framework can naturally handle sequential evidence collection and multi-turn interactions.
>
> We apologize for not making this explicit in the original manuscript. In the revised version, we include a **representative three-turn diagnosis example** in **Appendix A.12**, where additional results are split into IHC and molecular testing stages. The diagnosis becomes increasingly specific, evolving from “lymphoma (especially bronchial lymphoma)” → “B-cell-derived lymphoma (particularly MALT lymphoma)” → final diagnosis: “MALT lymphoma.”
>
> ---
>
> **W2. On reliance on LLM judges for both training and evaluation. Human evaluation matters.**
>
> We thank the reviewer for raising this concern. During training, we used a moderately capable open-source model (**Qwen2.5-32B-Instruct**) to compute rewards. For evaluation, we employed three stronger models (**Qwen-Max, Gemini2.5-Pro, GPT-5**) to assess differential diagnosis accuracy, final diagnosis accuracy, and examination plausibility. Thus, the observed improvements cannot be solely explained by alignment with a single judge or reward hacking.
>
> We fully acknowledge the importance of human evaluation and added this as a future direction in the discussion (**lines 529–532**). Due to practical constraints in medical studies, multi-model evaluation is a reasonable **proxy** for preliminary validation. In particular:
>
> 1. Prior studies show high agreement between LLM-based and human evaluations, and our use of three independent models further mitigates potential bias.
> 2. Full-scale human evaluation in medical settings requires substantial resources, which is challenging for an early-stage exploratory study.
>
> ---
>
> **W3. On clinical validation of examination simulation (RAGES).**
>
> Thank you for this comment. We acknowledge that clinical validation is ideal but infeasible at this early stage. To reduce concerns about model-to-model agreement, we applied different models for generating and evaluating examinations (**DeepSeek-R1 and GPT-5**). Additionally, to improve reliability, simulated examinations integrate: (1) **original real outcomes**, (2) **structured knowledge from our database**, and (3) **the large model’s intrinsic knowledge**. Our results indicate that this approach generates **plausible and useful simulated examinations**.
>
> ---
>
> We are very grateful for your careful review. Your comments help us clarify the framework’s core concept, present the ideas more clearly, and highlight the importance of human doctor validation in future work.

---

### Author Response · Authors · 2025-12-03
**A Rebuttal Summary for AC (Authors' View) --- Part 1**

Dear AC,

We all know that OpenReview has gone through a difficult period recently. To help minimize the additional reviewing burden and also take this opportunity to concisely summarize the major issues raised by reviewers along with our responses, we provide the following brief summary.

---

We would like to first briefly highlight the core contributions of our work:

- To the best of our knowledge, we are the first medical reasoning LLM trained via RL to tackle **inherent ambiguous diagnoses** with **active information acquisition**. Here, the inherent ambiguity arises from the insufficiency of clinical evidence, which can be gradually solved through specific examinations. We design a reinforcement learning framework with **verifiable rewards** adapted for this ambiguous diagnosis setting. The reward consists of three components: **format reward**, **rank-sensitive diagnosis reward**, and **examination consistency reward**. The rank-sensitive reward encourages the model to produce **a ranked list of possible diseases**, aligning with human experts' differential diagnosis. We **theoretically** prove that this reward favors **earlier hits** and **shorter differential lists**, yielding more accurate diagnoses. Moreover, the **dynamic tau mechanism** balances exploration (broad differential coverage) and exploitation (focused primary diagnosis).
- We formalize diagnosis as a **clinically grounded, evidence-seeking multi-turn process**, moving beyond single-turn evaluations. Empirical results show that LLMs achieve **substantially better diagnostic performance** when allowed to actively gather additional evidence, mirroring real clinical reasoning and demonstrating the model’s capability to identify when further information is needed for precise conclusions.

- To complete the evidence-seeking loop, we introduce **RAG-based Examination Simulation (RAGES)**. RAGES integrates real documented test results, a structured disease-testing knowledge base, and knowledge from powerful LLMs. It can generate **plausible and medically grounded** examination outcomes upon the model's request. Ablation studies show that RAGES outperforms pure generative approaches.
- We evaluate our approach on **multilingual datasets of public pathological cases**, where evidence is clearly separated into initial microscopic findings and follow-up test results. Our RL-trained model (Qwen2.5-7B-Instruct) demonstrates **competitive performance against multiple baselines**, including Qwen2.5-32B, Qwen2.5-72B, QwQ-32B, Qwen3-32B, and Baichuan-M2-32B (medical reasoning model), while providing interpretable, evidence-driven diagnostic reasoning.

Then we provide a simple summary of the strengths stated by the reviewers. To provide a more objective statement, we ask LLMs to summarize (we have double-checked to prevent hallucination).

- Reviewers agreed that the paper presents a clear and rigorous formulation of multi-turn, evidence-seeking diagnosis, offering **a clinically meaningful shift** beyond the common single-turn paradigm.
- The reinforcement learning framework, including the rank-sensitive reward, dynamic tau strategy, and supporting format/consistency rewards, was praised as **theoretically grounded and thoughtfully designed**.
- Reviewers also highlighted **the novelty and careful construction** of the overall two-turn diagnostic architecture and RAGES, which combines real data, structured knowledge, and LLM generation to produce more reliable evidence.
- The paper is considered well written, and the experiments, spanning multilingual datasets, strong external LLM judges, and ablation studies, provide **comprehensive validation of the method’s effectiveness** over multiple baselines.

Below are the issues and our responses in the initial rebuttal stage.

---

**Common Problem 1: The proposed method seems limited to a two-turn paradigm. (Reviewers Wi1g and hXAC)**
**Our response:** The method is not inherently constrained to a two-turn format. In fact, we have added a new case in the revised manuscript that includes **three turns**, demonstrating progressively deeper diagnostic reasoning (**Appendix A.12**).

---

**Common Problem 2: The training time seems too long; analysis of training and inference efficiency is needed. (Reviewers RqfJ, rvSr, and hXAC)**
**Our response:** We provide a detailed analysis in **Appendix A.6**. The primary sources of high training costs are the OpenRLHF framework's inefficient GPU utilization with additional LLM judges at that time, and the substantial expense of LLM judges as Qwen2.5-32B.

To be continued in Part 2...

---

> ### Author Response · Authors · 2025-12-03
> **A Rebuttal Summary for AC (Authors' View) --- Part 2**
>
> **Common Problem 3: The Ours-RL-7B model does not show a clear advantage, especially in DxAcc. (Reviewers RqfJ and rvSr)**
> **Our response:** Motivated by RqfJ, we additionally evaluate **HuatuoGPT-o1-7B** (a medical reasoning LLM based also on Qwen2.5-7B) and Ours-SFT-7B, medical-related reasoning models with similar sizes to Ours-RL-7B. The results (shown below) clearly demonstrate the superiority of our RL model. More analysis is in **Appendix A.5**.
>
> | Model       | DiffAcc (EN) | DiffAcc (CN) | DiffAcc (Mean) | DxAcc (EN) | DxAcc (CN) | DxAcc (Mean) |
> | ----------- | ------------ | ------------ | -------------- | ---------- | ---------- | ------------ |
> | Huatuo-7B   | 31.5         | 40.1         | 37.7           | 19.7       | 22.5       | 21.7         |
> | Ours-SFT-7B | 45.5         | 51.2         | 49.6           | 29.4       | 30.4       | 30.1         |
> | Ours-RL-7B  | 62.0         | 65.8         | 64.8           | 36.4       | 39.0       | 38.2         |
>
> **(New Reply)** Regarding applying RL to larger models, this is a substantial workload. After our initial responses, we urgently sought computing resources and attempted to train Qwen2.5-32B-Instruct. After multiple trials, we managed to begin training with 16x H200 GPUs (RL) and 8x H100 GPUs (LLM judge). The training was expected to take over 120 hours due to slower rollout generation and inter-machine communication. Unfortunately, training only reached about 30%, so we cannot provide final evaluation results. However, we observed that early checkpoints already outperform the base model, suggesting strong potential:
>
> | Model       | DiffAcc (EN) | DiffAcc (CN) | DiffAcc (Mean) | DxAcc (EN) | DxAcc (CN) | DxAcc (Mean) |
> | ----------- | ------------ | ------------ | -------------- | ---------- | ---------- | ------------ |
> | Qwen-32B    | 42.7         | 51.9         | 49.3           | 24.5       | 34.2       | 31.4         |
> | Ours-RL-32B | 45.8         | 60.1         | 56.0           | 32.1       | 43.9       | 40.5         |
>
> ---
>
> **Common Problem 4: Generalization of the model. (Reviewers hXAC and rvSr)**
> **Our response:** Inspired by hXAC, we compare model performance on the widely used MedQA dataset in **Appendix A.4**. The results show no significant degradation, but also no substantial improvement, likely because our training scope focuses specifically on diagnosis using only 900+ cases.
>
> | Model        | English (1273) | Simplified Chinese (3426) |
> | ------------ | -------------- | ------------------------- |
> | Qwen2.5-7B   | 53.5           | 82.8                      |
> | Qwen2.5-32B  | 68.5           | 89.7                      |
> | Ours-RL-7B   | 52.5           | 82.9                      |
> | Ours-SFT-32B | 74.0           | 87.1                      |
>
> ---
>
> **Common Problem 5: Human experts should be involved. (Reviewers Wi1g and rvSr)**
> **Our response:** Human evaluation is important but time- and resource-intensive. Therefore, we employ strong LLMs for evaluation, which have demonstrated high consistency with human doctors in Google Research’s AMIE. We also use three external LLMs (GPT-5, DeepSeek-R1, Qwen-Max) that are **not involved in training** to ensure robustness. This trade-off does not diminish the value of human experts. As discussed in lines 529-532, experts remain essential in medical model development, particularly for evaluation and safety alignment.
>
> ---
>
> **Common Problem 6: Presentation. (Reviewers rvSr and hXAC)**
> **Our response:** We have added an illustration of our rank-sensitive diagnostic reward in **Fig. 3**, and we rewrote the introductory part of the method section (**Section 3.1**) to provide a clearer overview of the framework.
>
> ---
>
> **Problem 7 (Reviewer RqfJ): Need evidence that two-turn diagnosis is better than one-turn.**
> **Our response:** In **Section 5.1** (“More Evidence, More Accurate Diagnosis”), we show that diagnostic accuracy consistently improves with additional evidence. To further support this, we conducted an idealized experiment during the rebuttal, providing information unavailable in the first turn (**Appendix A.3**). The main findings are:
>
> - Evidence-seeking diagnosis improves accuracy for most models.
> - Differential accuracy reveals substantial improvement space, highlighting the importance of sequential evidence acquisition.
>
> ---
>
> **Problem 8 (Reviewer rvSr): Dynamic integration of RAGES.**
> **Our response:** Integrating RAGES into an online loop is an important direction for the future, as discussed in the original paper. However, online RAGES would require significantly more resources than current judges. We tend to switch to the VeRL, which is more friendly with this tool in a call-like fashion.
>
> ---
>
> In the end, we sincerely thank all reviewers for their valuable feedback, which has significantly improved our paper. We especially thank Review RqfJ for raising the score to 6 before "that event". We also thank the AC for your patience in reviewing these issues and our responses.

---

### Note · Authors · 2026-01-27

I have read and agree with the venue's withdrawal policy on behalf of myself and my co-authors.

---

### Meta-Review · Area_Chair_FxRn · 2026-01-07

**Summary:**

Remaining concerns center on a lack of human assessments and quality check, computational inefficiency, and the real-world impacts when larger base models – the size of which is often more tolerated for high-stake decisions -- can still surpass carefully tuned smaller models. Other concerns such as limitations of the original two-round setup, additional evaluations on generalization, and issues in presentation have been generally addressed in the rebuttal.

**Reviewer Concerns:**

`Wi1g`: Failure to cover the more general multi-turn scenario (addressed with 3-turn experiments), reliance on LLM-as-judge and lack of human assessments (acknowledged practical constraints), quality of simulation lacks realism check (responded with surrogate experiments on model-to-model agreements).

`RqfJ`: Failure to cover the more general multi-turn scenario (addressed with 3-turn experiments); heavy computation (acknowledged); Concerns over effectiveness second turn dialogue for DxAcc of Ours-RL-7B in Table 2 (addressed with additional experiments); suggested for additional single-turn baselines; comparisons with other models (HuatuoGPT-o1); Reviewer acknowledged that most concerns had been solved and a score raise was promised before the platform incident.

`rvSr`: Performance behind larger reasoners in the context of high-stake decisions (remains outstanding, authors provided evidence of performance gain with models at similar sizes); lack of real-time interaction in training (limitation acknowledged); reward interpretation (addressed with additional materials), generalization (addressed with additional experiments on MedQA). Reviewer followed up mentioning that many raised concerns fall beyond the reach of a rebuttal and insist on the original scores.

`hXAC`: Limitations of two-round setup (addressed with experiments on three-round setups); generalization to more datasets (addressed with experiments on MedQA); Presentation, details on computing bottleneck in training (revised, additional information supplemented); reviewer decided to remain the score due to formulation and computational inefficiency.

**Reviewer Scores:**

The last three reviewers have made clear follow-ups before the platform incidence. For `Wi1g`’s concern over lack of human assessment and quality check over realism of simulation, the authors acknowledged this is a practical constraint. They provided a cross-model agreement experiment as a surrogate to realism check.

---

### Decision · Program_Chairs · 2026-01-26

Reject